# METATT: A GLOBAL TENSOR-TRAIN ADAPTER FOR PARAMETER-EFFICIENT FINE-TUNING

## ABSTRACT

We present MetaTT, a Tensor Train (TT) adapter framework for fine-tuning of pre-trained transformers. MetaTT enables flexible and parameter-efficient model adaptation by using a single shared TT to factorize transformer sub-modules. This factorization indexes key structural dimensions, including layer and matrix type, and can optionally incorporate heads and tasks. This design allows MetaTT's parameter count to scale with the sum, rather than the product, of the modes, resulting in a substantially more compact adapter. Our benchmarks compare MetaTT with LoRA along with recent state-of-the-art matrix and tensor decomposition based fine-tuning methods. We observe that when tested on single-task standard language modeling benchmarks, MetaTT achieves competitive parameter efficiency to accuracy tradeoff. We further demonstrate that MetaTT performs competitively when compared to state-of-the-art methods on multi-task learning. Finally, we leverage the TT-ansatz to design a rank adaptive optimizer inspired by the DMRG method from many-body physics. Our results demonstrate that integrating this approach with AdamW enhances optimization performance for a specified target rank.

## 1 INTRODUCTION

The sheer size of today's pre-trained models (e.g., LLaMA-3 Grattafiori et al. (2024), Gemini-1.5 Team et al. (2024), GPT-4o Hurst et al. (2024), Falcon-40B Almazrouei et al. (2023), Mistral-7B Jiang et al. (2023), BERT Devlin et al. (2019)) coupled with the rapid rise of adapting them for specific tasks has proven to be a catalyst for the research on parameter efficient fine-tuning (PEFT) methods. Since Aghajanyan et al. (2020) demonstrated that pre-trained language models can effectively learn on a given task even when subjected to a random projection onto a smaller subspace, starting from LoRA Hu et al. (2021), a flurry of works on PEFT have demonstrated significant parameter reduction for fine-tuning large models on simpler and often single tasks rev: Karimi Mahabadi et al. (2021a); Zhang et al. (2023); Zi et al. (2023); Zhang & Pilanci (2024); Albert et al. (2025).

To push beyond local layer-wise parameter reduction, sharing low-rank adapters across transformer layers have shown great promise. VeRA Kopiczko et al. (2024) shares single pair of low-rank matrices across all layers and learns small scaling vectors; NOLA Koohpayegani et al. (2023) re-parameterizes low-rank matrices as linear combinations of random bases, decoupling parameter count from rank and architecture; and VB-LoRA Li et al. (2024) and Uni-LoRA Li et al. (2025) construct all adapters from a global vector bank, achieving extreme parameter efficiency.

Subsequently, a wider class of methods have started considering the weight matrices (individually or a combination of them) as higher order tensors and then designing decompositions. To this end, several lines of work have emerged, partly because it is not obvious which components of the model benefit from tensor decompositions, and partly because unlike matrices, it is not known how to decompose tensors optimally Kolda & Bader (2009). As a result it remains open, even empirically, to understand how tensor-based decompositions can further improve the balance between the number of trainable parameters and downstream task performance during fine-tuning.

Methods compressing per layer adapters via tensor decompositions include LoRETTA Yang et al. (2024), which at each layer replaces LoRA's trainable matrices with a tensor train (TT) decomposition; TT-LoRA Anjum et al. (2024), which similarly first folds each trainable matrix into a tensor and then

factors them into TT decomposition. QuanTA Chen et al. (2024) and Quantum-PEFT Koike-Akino et al. (2025) further decompose adapters into tensor networks shaped as quantum circuits.

A promising avenue is to exploit *both* shared adapters and tensor decompositions, which can a priori lead to higher compression rates compared to per-layer tensor decompositions, at higher expressibility compared to shared low-rank matrices. In FacT Jie & Deng (2023), the authors use 3D TT and Tucker decompositions to capture parameter sharing across layers in the context of vision transformers. This idea is extended in the context of LLMs in LoTR Bershatsky et al. (2024).

Finally, LoRTA Hounie et al. (2024) decomposes the various linear layers in a transformer using a CP-decomposition. CP decompositions for higher order tensors generally have unique decompositions Kolda & Bader (2009), and thus finding the right decomposition may seem to be harder during fine-tuning. Given the plethora of work on matrix and tensor decomposition based adapters we ask,

*Can we achieve further parameter efficiency when fine-tuning transformer models?*

**A global adapter via tensor trains:** Building on LoTR and FacT-TT, which stack and decompose all adapter layers into a single 3D TT, we further separate the layer and matrix type dimensions to form a 4D TT, enabling greater parameter reduction. Decomposing the multi-head self-attention (MHSA) output into head and number of heads yields a 5D tensor, as in LoRTA. Both 4D and 5D TTs balance parameter efficiency and expressivity, unlike Tucker decomposition, which scales exponentially with dimensions, and is often more expressive than CP Khrulkov et al. (2017).

In the context of adapter-based methods, single-task fine-tuning has been widely studied. Only recently the need for multi-task learning (MTL) has gained prominence. This is partly due to the size of the pre-trained models and partly due to the fact that often there are common modalities across datasets and tasks. Modifications to LoRA has been shown to work well for MTL, e.g., ensembling multiple LoRA adapters Wang et al. (2023), using a mixture of experts (MoE-LoRA) Liu et al. (2024), and sharing one parameter across multiple tasks (MTL-LoRA) Yang et al. (2025). We observe that a tensor based structure automatically extends to MTL and so given a tensor based adapter one can construct a unified adapter, which may lead to further parameter efficiency rev: via parameter sharing across tasks, which has been shown to yield benefits in the setting of hypernetwork adapters Karimi Mahabadi et al. (2021b).

**Multi-task learning via tensor trains:** For MTL, the resultant tensor of the combined adapters of a pre-trained model, assuming task-specific adapters, has an extra dimension from labeling different tasks. As such, one can efficiently extend the family of TT adapters for MTL. Owing to the modular design of the TT-based adapter, we refer to this unified family of TTs as *MetaTT*.

Since the construction of such *global* adapters can also be achieved efficiently by means of other tensor decompositions (such as CP), it becomes natural to ask,

*Do we achieve anything beyond parameter reduction when using MetaTT's architecture?*

To address this, we examine the unique optimization advantages offered by the TT structure.

**Rank adaptive training:** Unlike other tensor decompositions, TTs are equipped with powerful optimization routines that exploit the TT structure. Specifically, we apply a rank adaptive scheme inspired by the Density-Matrix Renormalization Group (DMRG) optimization Schollwöck (2011); Verstraete et al. (2023), a method widely used in the context of quantum many-body physics, to improve training in the presence of many TT cores and adaptively choose the TT ranks during fine-tuning. Such a method does not trivially extend beyond the TT architecture.

## 2 META-ADAPTER WITH TENSOR NETWORKS

Tensor networks are mathematical structures that can represent high-dimensional tensors in a more manageable form. This is achieved by decomposing a large order tensor into a network of interconnected, generally lower-dimensional, tensors. This decomposition reduces the storage and computational requirements, making tensor networks suitable for applications involving big data.

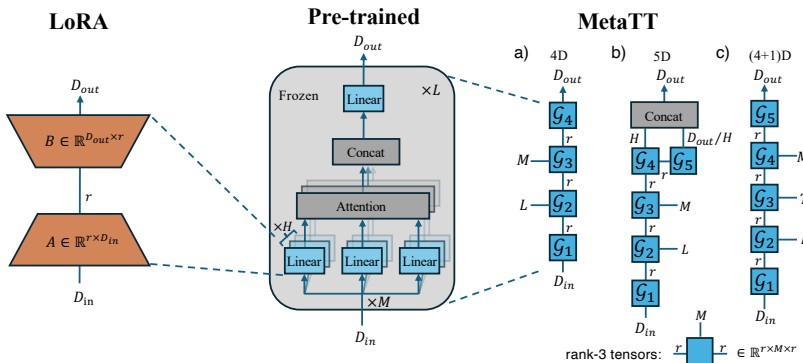

Figure 1: **Comparison between LoRA and MetaTT adapters.** rev: While LoRA parameterizes each weight matrix individually, MetaTT parameterizes all linear maps in the transformer architecture jointly as a TT (here shown only for a MHSA block). We propose two architectures for single-task fine-tuning: a) MetaTT-4D decomposes the entire set of linear maps into a TT of order 4 along the input/output dimensions (as in LoRA) as well as along the layer dimension, $L$, and the set of projection matrices, $M$. b) MetaTT-5D further decomposes the output dimension along the head dimension and number of heads. To capture task dependencies in multi-task learning, we extend MetaTT by adding an additional tensor core with mode dimension $T$ corresponding to the number of tasks, resulting in c) MetaTT-(4+1)D. Unlike LoRA, TT ranks in MetaTT can adapt during fine-tuning, providing both parameter efficiency and optimization flexibility.

## 2.1 TENSOR-TRAIN DECOMPOSITION

Among the various types of tensor networks, tensor trains (TTs) offer a particularly efficient representation. A TT decomposes a tensor $\mathcal{G} \in \mathbb{R}^{n_1 \times \cdots \times n_d}$ of order $d$ into a set of rank-3 tensors as follows

$$\mathcal{G}[i_1, \cdots, i_d] = \mathcal{G}_1[i_1]\mathcal{G}_2[i_2] \cdots \mathcal{G}_d[i_d], \tag{1}$$

where $\mathcal{G}_k[i_k] \in \mathbb{R}^{r_{k-1} \times r_k}$, $i_k = 1, \cdots, n_k$, are matrices, except the first and last, which are row and column vectors, respectively. It is also customary to see $\mathcal{G}_{k=2,\cdots,d-1}$ as rank-3 tensors, also known as cores. The parameters $r_i$ are known as TT-ranks. The complexity of the TT ansatz is $\mathcal{O}(dr^2n)$ with $r = \max_k r_k$, $n = \max_k n_k$. A TT therefore offers a controllable trade-off between expressivity (via $r$) and storage. In what follows, we assume the TT-ranks are all of equal value $r$.

## 2.2 TENSOR BASED ADAPTERS

LoRA type adapters inject a matrix $\Delta W \in \mathbb{R}^{D_{\text{in}} \times D_{\text{out}}}$ at every layer and a subset of projection matrices in the MHSA module. rev: The set of all such adapters can be viewed as a 4-dimensional tensor,

$$\Delta \mathcal{W}_{4D} = \{\{\Delta W_{l,m}\}_{l=1}^L\}_{m=1}^M \in \mathbb{R}^{D_{\text{in}} \times L \times M \times D_{\text{out}}}, \tag{2}$$

where $L$ is the number of layers, and $M$ is the number of projection matrices, which can be between 1 and 4 (corresponding to $Q, K, V$, and $O$ matrices). One can choose to include the MLP matrices in this tensor after properly reshaping them. For instance in the BERT family of models, the two MLP matrices are of size $4D_{\text{in}} \times D_{\text{out}}$. However, including these MLP layers would increase $M$ to potentially 12. This puts a significant computational overhead should we wish to construct a 3-dimensional tensor by stacking the $L$-dimension on top of $M$. In Jie & Deng (2023), the authors follow such an approach in the context of vision transformers to construct unified tensors to inject into the model.

Moreover, the output dimension in the MHSA is further split into $H$ number of heads. Thus implicitly, such a neural architecture allows for the construction of even a 5-dimensional tensor,

$$\Delta \mathcal{W}_{5D} = \{\{\{\Delta W_{l,m,h}\}_{l=1}^L\}_{m=1}^M\}_{h=1}^H \in \mathbb{R}^{D_{\text{in}} \times L \times M \times H \times D_{\text{out}}/H}. \tag{3}$$

Tensors $\Delta \mathcal{W}_{4D}$ and $\Delta \mathcal{W}_{5D}$ potentially capture all adapters one could include for a transformer-like model. Furthermore, this idea extends beyond single transformer adapters. For instance, one could

include an extra dimension capturing task dependency in a MTL (MTL) setting, where different transformer adapters are used for different tasks

$$\Delta \mathcal{W}_{6D} = \{\{\{\{\Delta W_{l,m,h,t}\}_{l=1}^{L}\}_{m=1}^{M}\}_{h=1}^{H}\}_{t=1}^{T} \in \mathbb{R}^{D_{\text{in}} \times L \times M \times H \times T \times D_{\text{out}}/H}, \tag{4}$$

where $T$ represents the total number of tasks.

## 2.3 MetaTT Adapter

Consider the TT decomposition of $\Delta \mathcal{W}_{4D}$, which we refer to as MetaTT-4D, shown pictorially in Figure 1. For fixed layer $l$ and $m$-th projection matrix, the TT decomposition results in a list of 4 matrix multiplications of rank $r$ (assuming for simplicity each bond has the same fixed rank). It is low-rank if $r \ll \min\{D_{\text{in}}, D_{\text{out}}\}$. In other words, for an input batch $X \in \mathbb{R}^{N \times D_{\text{in}}}$ and output batch $Y \in \mathbb{R}^{N \times D_{\text{out}}}$, for every layer $l$ and $m$-th projection matrix we have:

$$\begin{aligned} Y &= X \cdot W_{l,m}^{T} + \alpha X \cdot \text{TT}(\Delta \mathcal{W}_{4D})_{l,m} \\ &= X \cdot W_{l,m}^{T} + \alpha X \cdot \mathcal{G}_1 \mathcal{G}_2[l] \mathcal{G}_3[m] \mathcal{G}_4, \end{aligned} \tag{5}$$

where $W_{l,m}^{T}$ is the transposed frozen linear layer (from the pre-trained model). While one can choose any permutations of the ordering of the TT cores, we present the arrangement that leads to the most compressed form. This entails assigning the input and output dimensions to each end of the TT since these are usually the largest dimensions in a transformer architecture. This is because they are only coupled by a single bond, thereby incurring an additive $O(D \times r)$ cost to the overall complexity while also being quadratic in $r$ for the other dimensions, which are usually orders of magnitude less than the input and output dimensions.

The extension to capture other dimensions for the 5D variation with mild modifications is straightforward. One would need to concatenate the number of heads and the head dimension into the output dimension. This is shown in Figure 1. Importantly, minimal reshaping is required throughout this process, as shown explicitly for MetaTT-4D in equation 5. This is in contrast to other tensor decompositions. The input data is processed in its original format and outputs dimensions are also consistent with the original model, facilitating the use of optimized matrix-vector GPU kernels and allowing for performance and scalability enhancement. While further compression can be achieved by further unrolling the input and output dimensions, it is crucial to avoid these, as they can complicate the decomposition and reduce computational efficiency Monturiol et al. (2025); Lu et al. (2025).

**Complexity Analysis.** MetaTT-4D has $2Dr + (L + M)r^2$ parameters for $D = \max\{D_{\text{in}}, D_{\text{out}}\}$. Similarly, MetaTT-5D has $(D + D/H)r + (L + M + H)r^2$ parameters. This is substantially better than the LoRA adapter which requires at least $2LMDr$ parameters. Thus, by introducing a small series of $r \times r$ matrices we are able to significantly compress the tensor otherwise obtained by using LoRA. Note that for fixed TT-rank $r$, MetaTT-4D is more efficient than MetaTT-5D whenever $r > D/H(1 - 1/H)$.

Training times of TT adapters are very competitive with LoRA. At each linear layer one performs $2(D \times r) + 2(r \times r)$ matrix multiplications, where the complexity is dominated by $(D \times r)$ since $D \gg r$. As such the total time required to train the adapter is very similar to that of a LoRA adapter. During inference, one can match the speeds of LoRA by adding a single pre-computation step where one can merge the middle tensor cores with $\mathcal{G}_1$ or $\mathcal{G}_4$ (for MetaTT-4D) once the adapters are trained.

## 2.4 DMRG-Inspired Sweep: A Rank Adaptive Training Algorithm

While the gold standard of training PEFT adapters has been gradient descent using optimizers like Adam Kingma & Ba (2014), these methods fail to take advantage of the tensor decomposition structure. For matrices and other small order tensor decompositions, Adam works remarkably well. However, for higher order tensor decompositions, e.g., tensor networks with many cores, training using gradient descent can be unstable Barratt et al. (2022). For instance, while MetaTT-5D can be more efficient than MetaTT-4D in terms of parameter count, training the latter can be more unstable. We propose the use of a rank adaptive scheme inspired by the DMRG method Schollwöck (2011); Verstraete et al. (2023), a variational algorithm widely used in quantum many-body physics to optimize TTs (also known as matrix product states in that context) representing quantum wavefunctions.

Starting with a sufficiently high-rank TT, we train with Adam for a few epochs and then apply a compression layer composed of a series of SVD decompositions on neighboring merged tensor cores and keep only vectors corresponding to the largest $r$ singular values. While we use full SVD decomposition one can use approximate SVD Halko et al. (2011); Musco & Musco (2015); Tropp & Webber (2023). Alternatively, more sophisticated importance scores can also be used to compute low-rank approximations Cohen et al. (2017); Musco & Musco (2017); Zhang et al. (2023). We successively compute these low-rank approximations until a desired rank is achieved. We state this procedure in Algorithm 1 (this particular version of the algorithm is analogous to a TT-rounding sweep but involving two sites as opposed to one). Note that since the ranks change after calling DMRG, and thus the number of trainable weights, one must reinitialize Adam moments after each truncation.

---

**Algorithm 1** DMRG-inspired sweep

---

**Input:** MetaTT $\text{TT}_{\text{dD}}(\Delta\mathcal{W})$ with ranks $r_0$, $d$ represents number of TT cores, target ranks $r$, and truncated SVD function tSVD.
1: **for** $i = 1$ to $d - 1$ **do**
2:     $M \leftarrow \text{MERGE}(\mathcal{G}_i, \mathcal{G}_{i+1})$               ▷ *Merge adjacent cores, reshape into matrix M*
3:     $U, S, V^T = \text{tSVD}(M; r)$                    ▷ *Rank r approximation using SVD*
4:     $\mathcal{G}_i \leftarrow U; \mathcal{G}_{i+1} \leftarrow SV^T$
5: **end for**
6: **for** $i = d$ to $2$ **do**
7:     $M \leftarrow \text{MERGE}(\mathcal{G}_{i-1}, \mathcal{G}_i)$            ▷ *Merge adjacent cores, reshape into matrix M*
8:     $U, S, V^T = \text{tSVD}(M; r)$                    ▷ *Rank r approximation using SVD*
9:     $\mathcal{G}_{i-1} \leftarrow US; \mathcal{G}_i \leftarrow V^T$
10: **end for**
**Output:** MetaTT with ranks $r$.

---

## 3 EXPERIMENTS

In this section, we perform three sets of experiments. In Section 3.1 we test the performance of MetaTT in the context of single-task fine-tuning against state-of-the-art methods. Our focus here is on commonsense reasoning tasks using the setup of Hu et al. (2023), and natural language understanding Wang et al. (2018). In Section 3.2 we compare the performance of MetaTT when adding an extra tensor for capturing task-specific knowledge in the context of MTL. Finally, in Section 3.3 we demonstrate that optimizing MetaTT using a variant of AdamW alternating with Algorithm 1 can further boost the performance of fine-tuning using MetaTT.

### 3.1 SINGLE-TASK FINE-TUNING

In this section we discuss the performance of various PEFT adapters along with MetaTT on single-task fine-tuning.

| | Method | Param$\times 10^5$ | ARC-c | ARC-e | BoolQ | HellaSwag | OBQA | PIQA | SIQA | WinoGrande | Avg |
|---|---|---|---|---|---|---|---|---|---|---|---|
| Llama-2-7b | Zero Shot | – | 46.5 | 74.5 | 74.7 | 75.9 | 47.0 | 78.8 | 46.1 | 69.5 | 64.1 |
| | LoRA (r=8) | 41.9 | **52.6** | 78.9 | **81.0** | 76.1 | 61.0 | 79.9 | 55.3 | 74.3 | 69.9 |
| | LoRA (r=16) | 83.9 | 51.8 | 77.9 | **78.9** | **76.7** | **70.0** | **80.3** | **56.1** | **76.6** | **71.0** |
| | VeRA (r=1024) | 3.27 | 48.3 | 76.9 | 74.7 | 76.2 | 52.2 | 78.5 | 47.8 | 70.0 | 65.6 |
| | LoTR (r=16) | 1.47 | 51.6 | **80.6** | 78.5 | 75.8 | 60.4 | 79.8 | 53.5 | 71.1 | 68.9 |
| | MetaTT-4D (r=16) | 1.40 | 50.9 | **79.2** | 77.2 | 75.6 | 63.4 | 79.5 | 51.0 | 71.4 | 68.5 |
| | MetaTT-4D (r=256) | 43.2 | **53.7** | 78.1 | 77.3 | **76.3** | 68.0 | 80.0 | 55.5 | 75.9 | 70.6 |
| Llama-2-13b | Zero Shot | – | 48.9 | 77.6 | 71.0 | 79.4 | 49.4 | 80.3 | 47.2 | 72.1 | 65.7 |
| | LoRA (r=8) | 65.5 | **57.3** | **81.6** | 82.4 | 78.8 | **62.0** | **81.3** | 53.8 | **76.2** | 71.7 |
| | LoRA (r=32) | 262.1 | **57.6** | 80.2 | **84.4** | **78.9** | 60.0 | 81.7 | **57.2** | **79.6** | **72.5** |
| | VeRA (r=256) | 4.3 | 53.2 | 79.8 | 80.3 | 77.8 | 57.4 | 80.8 | 49.6 | 74.4 | 69.2 |
| | LoTR (r=64) | 9.83 | 55.2 | 80.4 | 82.9 | 78.9 | 56.8 | 81.2 | 53.8 | 74.7 | 70.5 |
| | MetaTT-4D (r=16) | 1.75 | 55.0 | 80.6 | 83.4 | **79.2** | 55.2 | 81.1 | **54.5** | 75.1 | 70.5 |
| | MetaTT-4D (r=64) | 8.3 | 56.7 | **81.3** | **84.4** | 78.5 | **65.6** | 80.1 | 54.2 | 75.1 | **72.0** |

Table 1: **Comparison of fine-tuning Llama-2-7b and Llama-2-13b.** We show in bold the two best accuracies per task. We observe that MetaTT-4D trails very closely to LoRA while often outperforming VeRA while using $\approx$ 30x and $\approx$ 3x less trainable parameters respectively. We also observe that across both models, MetaTT and LoTR performs similarly, with slightly fewer parameters.

| | Method | Param $\times 10^3$ | Rank | Metric (%) | | | | | | | |
|---|---|---|---|---|---|---|---|---|---|---|---|
| | | | | CoLA | MNLI | MRPC | QNLI | QQP | RTE | SST2 | STS-B |
| RoBERTa_base | FT | 125k | – | 61(1) | 87.6 | 89.3(9) | 92.6(1) | 91.9 | 79(2) | 94.1(1) | 90.4(2) |
| | LoRA | 295 | 8 | **61.1(6)** | **87.3(2)** | 88(1) | 91.3(2) | **90.1(1)** | 73(2) | **94.2(2)** | 90.7(2) |
| | VeRA | 43 | 1024 | 58(1) | 81(3) | 87.2(7) | 89.6(4) | 85.87(2) | 73.4(9) | 92.2(4) | 88.7(4) |
| | LoRETTA_adp | 57 | 64,5[5] | 57.9(1) | 84.6(0) | 86.4(1) | 92.0(0) | 88.0(0) | 70.3(2) | 93.3(0) | –(–)[4] |
| | LoRTA | 6.9 | 8 | 55.9(1) | 84.1(0) | 86.9(1) | 91.1(1) | 86.7(0) | 70.2(1) | 93.0(1) | 86.6(0) |
| | LoRTA | 55 | 64 | 58.6(1) | **86.1(0)** | 88.0(2) | **92.2(0)** | **89.0(0)** | 75.0(2) | 93.6(0) | 89.3(0) |
| | LoTR | 74 | 32 | 60.5(?)[1] | 85.2(6) | 85.9(4) | 90.0(1) | 87.4(1) | 66(4) | 93.0(4) | 88.8(4) |
| | | 100 | 40 | 58(2) | 85.2(2) | 88(1) | **92.5(3)** | 87.6(0) | 53(14) | 93.8(7) | 89.8(5) |
| | | 276 | 80 | 61(2) | 84.6(1) | **89.0(0)** | 92.1(5) | 86.8(0) | 71(3) | 93.4(1) | **90.9(2)** |
| | | 321 | 88 | **61.3(6)** | 84.7(0) | 88.0(9) | 92.0(4) | 86.9(0) | 67(13) | 93.3(2) | **91.0(1)** |
| | MetaTT-4D | 13 | 8 | 58.8(5) | 84.2(1) | 87.6(2) | 90.4(1) | 86.9(1) | 72.9(5) | 92.0(1) | 89.1(2) |
| | | 45 | 24 | 59.7(7) | 85.5(1) | 88.6(4) | 91.0(1) | 87.5(1) | 74.2(4) | 92.3(2) | 89.9(2) |
| | | 156 | 64 | 61(1) | 85.9(1) | **88.9(3)** | 77(12)[2] | 88.5(1) | **77.5(7)** | 72(15)[2] | 90.1(2) |
| | MetaTT-5D | 20 | 16 | 50(2) | 84.0(1) | 88.2(5) | 89.7(1) | 87.0(1) | 73.6(8) | 93.2(3) | 88.6(3) |
| | | 160 | 64 | 60.4(3) | 85.8(1) | 88.8(2) | 91.3(2) | 88.3(1) | **74.8(8)** | **93.8(1)** | 89.5(4) |
| RoBERTa_large | FT | 355k | - | 68 | 90.2 | 91 | 94.7 | 92.2 | 87 | 96.4 | 92.4 |
| | LoRA | 786 | 8 | **68.0(7)[3]** | **90.6(2)** | 84(5)[3] | **94.8(3)** | **91.6(2)** | **87.0(8)[3]** | 95.7(2) | **91.9(4)** |
| | VeRA | 61 | 256 | 64(2) | 88.8(2) | 89.4(4) | 93.1(2) | 87.62(8) | 83(1) | 95.1(2) | 91.5(1) |
| | LoRETTA_adp | 133 | | 61.0(1) | 89.69(0) | 88.1(1) | 94.08(1) | 89.6(0) | 72.0(4) | 95.5(0) | –(–)[4] |
| | LoRTA | 9.1 | 8 | 58.9(1) | 88.4(0) | 87.3(1) | 94.0(0) | 88.1(1) | 66.6(10) | 95.3(0) | 91.1(0) |
| | LoTR | 328 | 64 | 61.3(9) | **90.3(0)** | 89.0(5) | **94.8(1)** | **89.2(1)** | 84(2) | **95.9(1)** | 91.6(1) |
| | MetaTT-4D | 39 | 16 | 62.8(5) | 89.6(1) | 88.6(3) | 93.8(1) | 88.5(1) | 84.2(5) | 95.2(2) | 91.8(1) |
| | | 92 | 32 | 64.0(1) | 90.0(1) | **90.1(3)** | 94.4(2) | 76(9)[2] | **84.8(6)** | 95.3(2) | **92.2(1)** |
| | MetaTT-5D | 78 | 32 | 63.2(5) | 89.8(1) | 89.6(1) | 93.4(0) | 88.7(1) | 73(7) | 94.6(0) | 91.5(2) |
| | | 242 | 64 | **64.9(2)** | 90.0(1) | **90.0(4)** | 93.4(1) | 89.1(1) | 74(9) | 95.2(1) | 65(23)[2] |

Table 2: **Comparison of MetaTT-4D and MetaTT-5D against other PEFT techniques on RoBERTa_base and RoBERTa_large.** Results for LoTR and LoRA are reported from Bershatsky et al. (2024). For each dataset, we highlight the two best PEFT methods (FT is not considered for this ranking and we only list it as a benchmark). For CoLA, the metric is Matthew's correlation, for STS-B it is the Spearman's rank-correlation coefficient, and for all other datasets it is accuracy. Observe that variants of MetaTT sometimes outperform or match the performance of LoRA for a much lower parameter count (between 20x and 2x less parameters when compared to LoRA). Value in parenthesis is a standard error rounded up to the last single significant digit.

**Commonsense reasoning.** We present results on the performance of MetaTT-4D against LoRA and other parameter-sharing methods introduced earlier in the context of commonsense reasoning: VeRA Kopiczko et al. (2024) and LoTR Bershatsky et al. (2024) in Table 1. We follow the same setup from Hu et al. (2023) and first train on the Commonsense170k dataset, and assess results across eight different downstream tasks. For fine-tuning, we utilize Llama-2 models Touvron et al. (2023) with 7B and 13B parameters as our pre-trained models. We report best accuracy results for each of the methods across two epochs (see Appendix D for more details on selection of hyper-parameters). For comparison we also include LoRA $r = 8$ and MetaTT-4D with $r = 16$. Due to the computational burden of the task and the models chosen, we report only single-shot results. Accuracies are evaluated using the lm-evaluation-harness framework Gao et al. (2024).

We observe that MetaTT-4D closely trails LoRA in terms of average performance across almost all the commonsense tasks for both Llama2-7b and Llama2-13b, while using significantly fewer parameters: up to $\approx 30x$ fewer parameters as compared to LoRA with less than $1\%$ drop in average accuracy. For both Llama2-7b and Llama2-13b, MetaTT-4D performs very similarly to LoTR while using fewer parameters and outperforms VeRA in almost all datasets while using $\approx 3x$ fewer trainable parameters.

**Language understanding.** We compare fine-tuning RoBERTa based models with MetaTT-4D and MetaTT-5D against several baseline methods on GLUE Benchmark datasets - CoLA, MNLI, MRPC, QNLI, QQP, RTE, SST2 and STS-B Wang et al. (2018) in Table 2. To isolate the performance of the shared adapters we only fine-tune the encoder adapter weights for the attention modules and not the classifier or regression heads for the corresponding downstream tasks. We defer the reader to Appendix D for a detailed exposition on hyper-parameter tuning, adapter target modules, and the final set of hyper-parameters used to produce Table 2.

Our results indicate that MetaTT is competitive with other state-of-the-art methods. It outperforms LoRETTA and VeRA. We also observe that irrespective of the rank, LoTR, LoRTA, and variants of MetaTT trail behind LoRA in accuracy across tasks, while using significantly fewer parameters. Finally, we reiterate that the accuracies reported in Table 2 were achieved by only fine-tuning the attention weights. We expect significant gains on top of these accuracies if the final classifier or regression heads are also trained. Note, the under-performance VeRA is expected when compared to higher order tensor decompositions, and is because the latter decompositions are able to affect a much larger number of parameters of the baseline model, given we fix the total number of trainable parameters. Among the LoRETTA variants, we find that the adapter-based method consistently outperforms the version that reparameterizes the input model. Therefore, we chose to report only the results for the adapter-based approach.

## 3.2 Multi-Task Learning

The modular architecture of TT based adapters allows for the inclusion of an extra tensor core capable of capturing task dependent features by simply assigning a low-rank rank-3 tensor along the TT chain. We explore this modification of the architecture by adding an extra core on MetaTT-4D placed at the middle of the TT, so that the ordering becomes $(D, L, T, M, D)$ where $T$ is the number of tasks on which the model is trained. We specifically choose this for symmetry of the tensor cores and not necessarily any particular reason. We compare this adapter, henceforth named as MetaTT-(4+1)D, against four baselines – a single LoRA adapter for all tasks and a MetaTT-4D adapter (which can be seen as a MetaTT-(4+1)D with the task dependent core frozen and set to identity), MTL-LoRA, and MoE-LoRA.

In the context of MTL, we distinguish between the following two approaches:

- **Sequential Learning.** This approach involves first fine-tuning a model on a specific task, transferring the adapter to a new task for further fine-tuning, and then transferring the adapter back to the original task. The core idea is to leverage the features learned from the second task to enhance performance on the first task. However, a significant challenge with sequential learning is the risk of catastrophic forgetting or training interference, where the model may lose previously acquired knowledge or experience negative interactions between tasks, respectively. These issues has been extensively studied and documented in the literature (see e.g., Zhang et al. (2024a)) and aligns with our observations.

- **Joint Training.** Alternatively, joint training aims to minimize a composite loss function that aggregates losses from multiple tasks at each epoch, i.e., the model is trained with the loss function $\mathcal{L} = \sum_{k=1}^{T} \mathcal{L}_k$, where $\mathcal{L}_k$ is the loss function for $k^{\text{th}}$ task.

**Experimental Setup.** Our experiments are again on fine-tuning RoBERTa$_{\text{base}}$ and RoBERTa$_{\text{large}}$ jointly on CoLA, MRPC, and RTE datasets from the GLUE benchmark datasets. A notable issue in

---

[1]The variance for this particular dataset and rank is not reported in Bershatsky et al. (2024).

[2]For specifically these runs, we found that the model does not train for values of alpha less than 1 (which were the ideal values found in the hyper-parameter search). We believe this is partly due to the initialization of tensor cores and partly because training tensor cores can be challenging (more on this later in Section 3.3). We defer further exposition on this to Appendix D.

[3]In LoTR Bershatsky et al. (2024) these values of LoRA failed to train successfully. For better comparison we re-run LoRA for these datasets using the same random seeds as for MetaTT. We found that on one run in MRPC LoRA failed to train successfully as well.

[4]We were unable find the right set of hyper-parameters for STS-B when freezing the final regression layers.

[5]In LoRETTA Yang et al. (2024), the bottleneck size is set as 64 for RoBERTa models and the TT rank is set as 5 for the adpater based method.

| Model | Method | Param $\times 10^3$ | Rank | Metric (%) | | | |
|---|---|---|---|---|---|---|---|
| | | | | CoLA | MRPC | RTE | Avg |
| RoBERTa$_{base}$ | LoRA | 295 | 8 | **60.7(8)** | 86.5(2) | **77.6(2)** | **74.9(2)** |
| | MTL-LoRA Yang et al. (2025) | 296 | 4 | 53.0(1) | **87.8(1)** | 71.6(2) | 70.8(2) |
| | MoE-LoRA Liu et al. (2024) | 307 | 8 | **60.1(1)** | **88.6(1)** | **82.8(3)** | **77.2(2)** |
| | MetaTT-4D | 13.2 | 8 | 53.2(2) | 85.9(4) | 72(2) | 70.3(8) |
| | MetaTT-(4+1)D | 13.4 | 8 | 54(1) | 86.0(5) | 71.5(5) | 70.5(8) |
| RoBERTa$_{large}$ | LoRA | 786 | 8 | **68(2)** | **89.3(6)** | **83.0(5)** | **80.0(3)** |
| | MTL-LoRA Yang et al. (2025) | 789 | 4 | 58.8(1) | 88.7(2) | 82.1(2) | 76.5(2) |
| | MoE-LoRA Liu et al. (2024) | 811 | 8 | 61.1(0) | **89.9(1)** | 82.6(1) | 77.8(1) |
| | MetaTT-4D | 18.0 | 8 | 59.5(5) | 88.4(5) | 81.1(8) | 76.3(6) |
| | MetaTT-(4+1)D | 18.2 | 8 | **64.0(8)** | 89.0(6) | **84.4(4)** | **79.2(4)** |

Table 3: **Results of MTL setup.** We observe that across both RoBERTa$_{base}$ and RoBERTa$_{large}$, MetaTT-(4+1)D outperforms single MetaTT-4D adapters for almost all of the datasets, while using about 200 more trainable parameters. For RoBERTa$_{base}$ MetaTT-(4+1)D performs comparably to MTL-LoRA while using $\approx 22\text{x}$ less parameters, and for RoBERTa$_{large}$ MetaTT-(4+1)D outperforms both MTL-LoRA and MoE-LoRA on average, and is within $1\%$ of average accuracy of LoRA, while using $\approx 43\text{x}$ less parameters. We show in bold the two best accuracies per task.

joint training is the disparity in dataset sizes, such as approximately 8000 training samples in CoLA compared to around 3000 in MRPC. To address this, we downsample each dataset to either the size of each dataset or a maximum of 5000 samples per dataset, whichever is smaller. This forms the training set. For the evaluation set, we retain either 500 samples or the full size of the validation/test set, whichever is smaller. For each trial, we first compute the mean performance across the three datasets at every epoch (yielding 20 points per trial) and then select the best mean among those 20 epochs. Finally, we report the average of these best means over 3 independent trials.

**Empirical results and observations.** The results from our experiments on MTL for 3 tasks are shown in Table 3. We first observe that a single LoRA adapter can work remarkably well across different datasets and pre-trained models. This had already been documented in (Yang et al., 2025, Table 1). For fine-tuning RoBERTa$_{base}$, we also observe that MoE-LoRA performs well across different datasets. While MetaTT-(4+1)D and MTL-LoRA perform similarly, MetaTT-(4+1)D uses about 13.4k parameters (an $\approx 22\text{x}$ parameter reduction when compared to other baseline methods). However, when we look at the performance on RoBERTa$_{large}$, we observe that MetaTT-(4+1)D performs within $1\%$ of LoRA, and outperforms both MoE-LoRA and MTL-LoRA, while requiring about 18.2k trainable parameters (an $\approx 43\text{x}$ parameter reduction when compared to other baseline methods). Furthermore, we observe that in case of RoBERTa$_{base}$, MetaTT-4D performs very similar to MetaTT-(4+1) while using about 200 less parameters, and in case of RoBERTa$_{large}$, MetaTT-4D performs very similar to MoE-LoRA and MTL-LoRA while using about 200 less parameters than MetaTT-(4+1)D (and using $\approx 43\text{x}$ less trainable parameters when compared to other methods). We defer further experiments and a discussion of how MetaTT-(4+1)D captures task dependent information for MTL to Appendix B.

### 3.3 RANK ADAPTIVE FINE-TUNING VIA DMRG-INSPIRED SWEEP

**Empirical evaluations.** We present comparisons of training RoBERTa$_{base}$ and RoBERTa$_{large}$ using AdamW Loshchilov & Hutter (2017) and interdispersing DMRG-inspired sweeps as in Algorithm 1 on the MRPC and RTE dataset in Figure 2. For AdamW Loshchilov & Hutter (2017) we fine-tune on fixed ranks $\{4, 6, 8\}$ for a given learning rate. We observe that one can adaptively change the ranks in the training phase without any major performance degradation. For RoBERTa$_{base}$, we show that using AdamW together with DMRG-inspired sweeps we achieve higher accuracy at rank $r = 4$ when compared to the accuracy achieved by AdamW and $r = 4$. We also show that the performance improvement with Algorithm 1 when fine-tuning RoBERTa$_{large}$ is even more significant (see Table 4). For both models, one can observe that the accuracy reduces significantly when truncated SVD is applied, followed by a rapid climb, with deeper gorges as we go to smaller ranks. Each DMRG sweep is applied right after each training epoch, before the evaluation on the validation dataset. Thus, it removes a significant amount of information, across all bonds of the TT, and so performance degradation is expected, before AdamW is able to readjust to its new weight space at the next epoch. This problem is exacerbated when DMRG is applied to smaller ranks, as the relative change in ranks

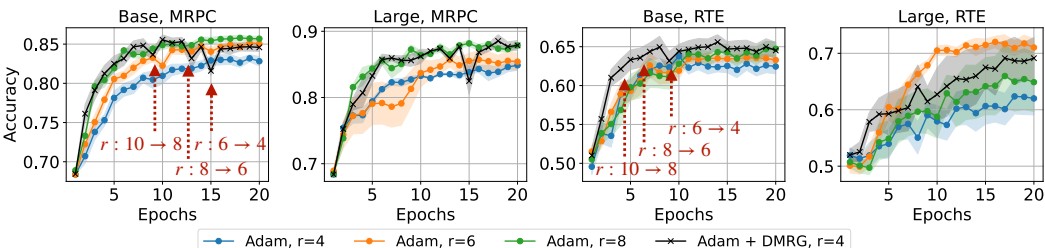

Figure 2: **Comparison of AdamW and AdamW+Algorithm 1 sweeps applied at certain epochs.**
Results are shown for MetaTT-5D on MRPC and RTE for RoBERTa$_{base}$ and RoBERTa$_{large}$. In Adam
we fix the rank throughout. For AdamW+Algorithm 1 we start with a $r = 10$ TT and progressively
decrease ranks until we reach $r = 4$ as indicated by arrows on the plots for the base model, with the
same schedule followed by the large counterparts. Error bars in both panels correspond to standard
errors. The learning rate used across all the optimizers is $5e - 4$ with $0$ weight decay.

(current rank divided by target rank) substantially increase as we go from higher ranks to lower ranks.
We defer further discussions on the experiments with DMRG to the Appendix C.

| Model | AdamW | AdamW + DMRG |
|---|---|---|
| Base, MRPC | 0.839 | 0.852 |
| Large, MRPC | 0.854 | 0.887 |
| Base, RTE | 0.652 | 0.658 |
| Large, RTE | 0.640 | 0.701 |

Table 4: Comparison of the average of per-trial maximum accuracies (computed over 20 epochs),
for 10 trials (RoBERTa$_{base}$) and 4 trials (RoBERTa$_{large}$), between AdamW and AdamW + DMRG
optimizers at target rank $r = 4$.

## 4    CONCLUSIONS

In this work, we have introduced MetaTT, a novel approach to parameter-efficient fine-tuning of
large language models using TT decompositions. By leveraging the TT architecture, MetaTT
achieves significant reductions in the number of trainable parameters while maintaining competitive
performance compared to state-of-the-art methods. Our empirical evaluations demonstrate that
MetaTT can achieve significant parameter reduction with similar accuracy on standard language
modeling benchmarks when compared to these methods.

The TT representation provides a compact and globally shared core, allowing for efficient parameter
sharing across all components of a transformer network. Unlike methods that compress each weight
matrix in isolation, MetaTT factorizes all linear sub-modules into a single shared TT, capturing
structural axes such as layer, matrix-type, and optionally heads and tasks. This global compression
leads to higher compression rates and improved scalability, making MetaTT a promising solution for
fine-tuning large models. However, for single task learning, we observe that MetaTT often performs
similar to other tensor based decompositions (including other variants of the TT decomposition).

To differentiate beyond single task fine-tuning, we observe that tensor based adapters can be easily
extended to perform joint-MTL. Notably, we demonstrate that extending a simple modification to the
architecture for single task learning, one can use MetaTT for joint-MTL. This is because the modular
architecture of MetaTT enables extension to shared adapters across multiple tasks or expert partitions,
without the need to redesign the core tensor. This had remained unexplored prior to our work. We
further hypothesize that similar extensions could be applied to other tensor-based architectures as
well. Furthermore, the TT ansatz benefits from mature optimization routines, such as DMRG-style
alternating minimization, which simplifies rank tuning. This allows MetaTT to adaptively choose TT
ranks, further enhancing its efficiency and performance.

Our results suggest that assuming low-rankness in the manifold of shared parameters is a viable strategy for parameter-efficient fine-tuning. The TT decomposition captures this manifold effectively, providing a robust framework for reducing computational overhead while preserving model performance. Future work may explore other tensor networks that better capture parameter sharing, including quantum-circuit inspired tensor network that may lift the low-rankness description while maintaining efficient parameter count.

While our focus here has been on fine-tuning, we anticipate MetaTT to find extensions to other contexts including the design of new foundation models with shared parameters and for model compression. Moreover, DMRG-inspired techniques can offer a principled way to compress TTs during training. Finding applications where compression during training phase or alternative DMRG-inspired techniques extending beyond those discussed in this work, presents an exciting avenue.

## 5 REPRODUCIBILITY STATEMENT

We provide all the codes and pseudocodes required for verifying experiments with MetaTT in Appendix E. Furthermore, the grids for hyper-parameter search and the final set of hyper-parameters required to reproduce the results of Table 1, Table 2, and Table 3 are reported in Appendix D and Appendix B. The experimental details for DMRG based experiments are reported in Section 3.3.

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

## A    OTHER RELATED WORKS

In this section, we explore various works that are pertinent to our study.

**Alternatives to tuning weights for PEFT.**    Among the relevant works, we briefly highlight research that investigates alternatives to tuning transformer weights for adapting to new tasks and/or datasets. Notably, few-shots in-context learning methods have been demonstrated to perform less effectively than PEFT methods Brown et al. (2020); Runwal et al. (2025). Alternative approaches, such as prompt tuning, aim to isolate and preserve the shared knowledge subspaces Zhang et al. (2024b); Khattak et al. (2023), while also learning context vector representations for prompts Zhou et al. (2022b;a); Zhang et al. (2024b). While these works significantly reduce the amount of learnable parameters, additional processing of input data can increase inference latency. Moreover, prompt tuning is often limiting beyond the realms of few-shots learning Han et al. (2024); Wistuba et al. (2024) and can be outperformed by appropriate low-rank fine-tuning in presence of more data Zanella & Ben Ayed (2024). As such similar to prior work Albert et al. (2025) on this area of parameter efficient fine-tuning, we compare our work to algorithms for weight tuning only.

**Tensor networks in machine learning.**    Early work on tensor networks were standalone models for supervised learning Novikov et al. (2016); Stoudenmire & Schwab (2016). Parallel to this, research exploited them for weight compression in convolution neural networks (CNNs) and recurrent neural networks (RNNs) Novikov et al. (2015); Garipov et al. (2016); Kim et al. (2015); Yin et al. (2022). These techniques have motivated much of the work around tensor networks for fine-tuning. We remark that while tensor networks can significantly compress individual neural-network (NN) layers, they present notable drawbacks in terms of computational efficiency and latency on GPUs due to the need to manage tensor contraction and reshaping Monturiol et al. (2025); Lu et al. (2025).

## B    FURTHER EXPERIMENTS ON MTL USING METATT-(4+1)D

In this section we provide further evidence that the task-related tensor cores in MetaTT-(4+1)D used in Section 3.2 play a significant role. For any given layer index $l$, matrix-type index $m$, and task index $t$, a given input batched vector gets updated as

$$X \leftarrow X \cdot W_{l,t,m}^T + \alpha X \cdot \mathcal{G}_1 \mathcal{G}_2[l] \mathcal{G}_3[t] \mathcal{G}_4[m] \mathcal{G}_5. \tag{6}$$

To show the impact of the inclusion of task-dependent TT cores, in Figure 3 we plot heatmaps of the gradients across each tensor in the TT. Since the boundary cores $\mathcal{G}_1$, $\mathcal{G}_5$ are much larger than the rest of the cores, we normalize the gradients across each TT core by the number of non-zero elements as follows $- ||\nabla_{\mathcal{G}}||_F / \sqrt{|\mathcal{G}|}$ where $|| \cdot ||_F$ is the Frobenius norm and $|\mathcal{G}|$ is the number of non-zero elements of the tensor $\mathcal{G}$. For tensors $\mathcal{G}_2$ and $\mathcal{G}_4$ we plot the average gradients across all layers and matrix-types, respectively. We observe that indeed the tensor $\mathcal{G}_3$ is acquiring significant gradient updates, especially for RoBERTa$_{large}$. Moreover, for certain epochs, we find that $\mathcal{G}_3$ in fact acquires the largest gradients across all tensors. Interestingly, we find that the task core with label 2 in Figure 3 corresponding to CoLA receives the largest gradient update. This is expected since CoLA is the hardest task among the chosen sets of tasks. Similar to Section 3.2, the rank chosen in these experiments is 8 across all bonds. Other relevant hyper-parameters used are: batch size $= 16$, $\alpha = 2$, learning rate $= 5e - 4$. Also similar to Section 3.2, we down-sample each dataset so as to contain a maximum of 5K training samples and 500 evaluation samples. Furthermore, we perform gradient clipping with a maximum gradient value of 3.0.

We compliment plots of gradients with the downstream task performance per epoch on each of the plots. While it is generally hard to make direct comparisons between gradients observed and downstream task performance, in Figure 3 we observe that for both RoBERTa$_{base}$ and RoBERTa$_{large}$ and the RTE dataset, the gradients observed at each epoch at tensor core $\mathcal{G}_3[1]$ correlate with the downstream task performance of the model.

## C    FURTHER DETAILS ON THE EXPERIMENTS WITH DMRG

The results of Section 3.3 show that, for a given target rank ($r = 4$ for both MRPC and RTE datasets), interspersing DMRG-inspired sweeps to progressively bring down the TT ranks from a high enough

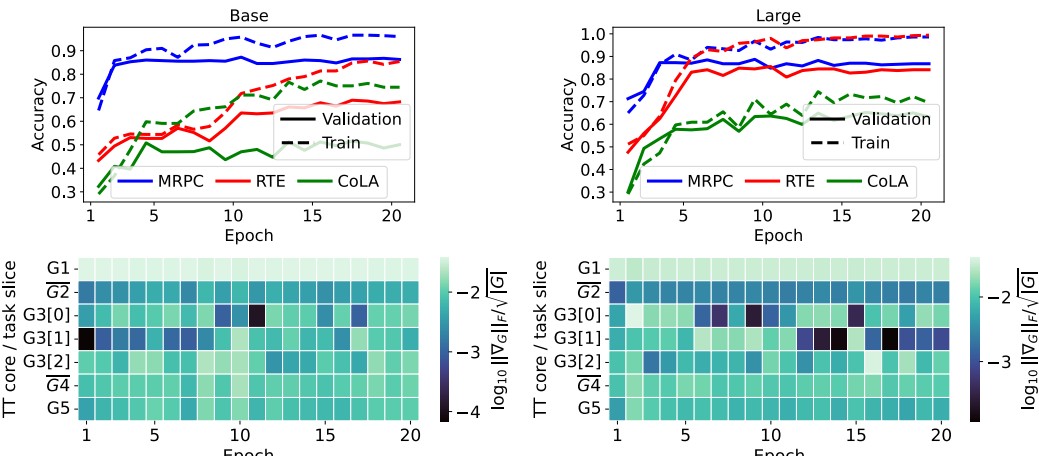

Figure 3: **Influence of task-dependent TT core in MTL.** (Left): (Top): Accuracy of MetaTT-(4+1)D as a function of epochs for RoBERTa$_\text{base}$ for a single training realization (in the case of CoLA we compute Matthew's correlation instead). (Bottom): Corresponding normalized gradients across all tensors as a function of epochs (see Appendix B). Task labels correspond to 0: MRPC, 1: RTE, 2: CoLA. (Right): Same as in left but for RoBERTa$_\text{large}$ as pretrained model.

rank ($r = 10$ in this case) leads to higher accuracies than training via AdamW with that fixed target rank. Interestingly, the rank schedule philosophy used here in DMRG is the mirror image of the one commonly used in many-body physics: there, one starts with small ranks and progressively increments these so as to capture more precisely the target *ground state* (see Schollwöck (2011) and references therein). Instead, in ML settings such as ours, the TT rank serves as a regularizer; pruning redundant directions after the optimiser has identified them improves generalization and reduces memory, whereas too high rank risks overfitting.

The choice of rank schedule was done heuristically, with the only consideration in mind that the ranks should be reduced slowly so that the model can adapt to the new weight space more efficiently. We see two potential extensions that find such rank schedules in more principled ways and that we leave open for future work.

First, for our experiments we used the magnitude of the singular values across TT bonds as diagnostic to shrink the ranks (even if they all remained high relative to each other). One improvement could come in the form of considering other *importance scores* that take into account the sensitivity of those singular values to the loss function. This would necessitate freezing all TT cores not involved in the SVD process. An approach similar in spirit was done in the context of LoRA type adapters in AdaLoRA Zhang et al. (2023). We remark here that one advantage of performing rank adaptive schemes based on SVDs in MetaTT over LoRA type adapters is that a much smaller fraction of SVDs are needed in MetaTT than in LoRA. This series of SVDs at the end of certain epochs result in a small overhead. This is in contrast to performing SVDs on all LoRA type adapters across the transformer architecture. It is for this reason that the orthogonality condition on the isometry factors stemming from SVDs are enforced through regularizers in AdaLoRA Zhang et al. (2023).

A second approach, which follows the original DMRG algorithm closer in spirit is to use powerful local optimizers to minimize directly the loss function with respect to each merged tensor at each step of the DMRG-inspired sweep in Algorithm 1. This would not only enable rank adaptation across each TT bond, but also directly optimize the loss function which may result in a powerful optimizer.

## D EXPERIMENTAL DETAILS

In this section we include experimental details not covered in the previous sections.

## D.1 Methodology for Hyper-parameter Search

During hyper-parameter tuning, we conducted a manual grid search without fixing random seeds, as the goal was to identify promising regions in the search space rather than produce final reportable results. The details of the grid search are given in Appendix D. For the final evaluation, we selected the best-performing configurations and ran them across three different, fixed random seeds (see Appendix D for the final set of hyper-parameters used) to ensure stability and reproducibility. This allowed us to balance exploration efficiency with reliable performance reporting. Generally, we follow Bershatsky et al. (2024) for hyper-parameter tuning. For very large datasets ($\geq$ 500k data-points, e.g., MNLI and QQP), we do the hyper-parameter tuning for only 1 epoch (for example MNLI which has $\approx$ 390k entries). For smaller datasets (e.g., CoLA and MRPC), we train for 20 epochs.

**Seeds.** We run 3 trials for most of our experiments unless the datasets are huge, in which case we run only 2 trails. All experiments with RoBERTa$_{\text{base}}$ use the following seeds $\{33305628, 2025, 42\}$, and the experiments with RoBERTa$_{\text{large}}$ use the following seeds $\{56346, 2025, 42\}$.

## D.2 Choice of Projection Matrices

All experimental results in the main text were obtained by adapting $Q, V$ matrices, as these were the ones used in LoTR Bershatsky et al. (2024), LoRA (Hu et al., 2021, Tables 2, 3), and VeRA Kopiczko et al. (2024). Just like several other PEFT adapters, MetaTT allows for fine-tuning any arbitrary subset of attention and projection matrices in a transformer architecture, including the MLP matrices (upon a proper reshaping). Since the number of projection matrices to be adapted $M$ (per layer) factorizes separately from other variables in MetaTT (including number of layers and input/output dimensions), higher compression rates can be achieved by considering this quantity larger. In line with previous works, we found that capturing $Q, K, V$ matrices at once did not improve over capturing only $Q, V$ matrices. We leave for future work a detailed study of the role of MLP layers and output projection matrices $O$.

## D.3 Implementation Environment

**Implementation details for MetaTT variants and other baselines.** To construct our training and benchmarking suite, we employed a range of technologies. HuggingFace provides a wrapper, known as HuggingFace Transformers Wolf et al. (2019), which extends existing deep learning libraries like PyTorch Paszke et al. (2019) with additional NLP functionalities. This library offers a unified interface for tasks such as input tokenization, model configuration, inference pipelines, and output decoding. We utilized HuggingFace's Transformers and PEFT Mangrulkar et al. (2022) to facilitate the design and training of our adapters, specifically taking advantage of the *Trainer* and *TrainingArguments* features available within the library.

**Implementation details for DMRG-inspired sweep.** Similar to the single task and multi-task learning, we leverage HuggingFace's transformers library Wolf et al. (2020) to load the models and HuggingFace datasets Lhoest et al. (2021) to load the datasets. However, we do not leverage the *Trainer* here and instead fall back to custom PyTorch training loops as we wish to have more precise control over the training loop (this is because we are changing the model itself during the run). Doing this using a custom PyTorch loop is much cleaner than using *TrainerCallbacks*.

**Machine configuration and coding environment.** We run our benchmarks on a machine with the following configuration: dual Intel Xeon Platinum 8275CL CPUs with 96 cores, 192 threads, and 1.1 TB of RAM and 8 A100 GPUs with 40GB memory each (an AWS P5 instance). At any given point on any GPU, only 1 model is being trained against one dataset.

## D.4 MetaTT

**Initialization.** An important component for running MetaTT successfully is the initialization strategy. There is freedom in choosing how to initialize each core, as long as the TT contraction $\mathcal{G}[i_1, \cdots, i_d] = 0$ along each slice. This is required to guarantee $\Delta W_{l,m} = 0$ everywhere at the beginning Hu et al. (2021). The majority of our experiments from the main text initialized the first

core $\mathcal{G}_1$ to zero, and the rest to the identity along each slice. I.e., $\mathcal{G}_i[j]_{k,l} = \delta_{k,l}$. This choice was done for simplicity and ease of reproducibility of results, and found to work well across all datasets that we experimented with. In Figure 4, we compare this scheme against other initialization strategies on MRPC and RTE. Note that further improvements on the numbers quoted in Table 2 can be achieved by optimizing initialization choices, as shown in Figure 4.

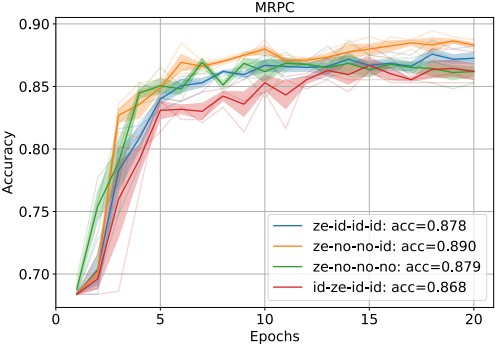 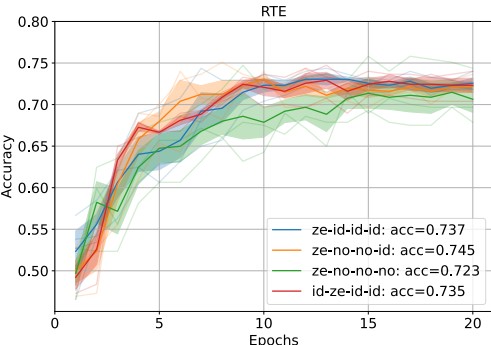

Figure 4: **TT initialization performance.** Shown are the accuracies in MRPC (left) and RTE (right) when training MetaTT-4D on RoBERTa$_{base}$ with different initialization strategies along with mean of best accuracies over 20 epochs across 3 different trials shown in the legend. Each pair of letters correspond to a different initialization strategy: 'ze' sets a given core to zero, 'id' sets each matrix slice of a core to the identity matrix and 'no' to a normal distribution with mean $= 0$ and standard deviation $= 0.2$. The order of pairs of letters follows the order of how each of the cores are initialized in MetaTT-4D. We choose the sequence ze-id-id-id (blue line) since it generally performs well on average across multiple datasets.

**Hyper-parameters for MetaTT results of Table 2.** In Table 5 and Table 6 we list the exhaustive set of hyper-parameters required to replicate the results in Table 2 for MetaTT-4D and MetaTT-5D respectively. For final evaluations, we run multiple trials across all the datasets for each transformer (for 20 epochs). For CoLA, MRPC, RTE and STS-B, we do 3 trials. For MLNI, QNLI, QQP and SST2, we run 2 trials due to their large cardinality.

| Model | Rank | Params | CoLA | MNLI | MRPC | QNLI | QQP | RTE | SST2 | STS-B |
|---|---|---|---|---|---|---|---|---|---|---|
| RoBERTa$_{Base}$ | 4 | $\alpha$ | 4 | 4 | 0.5 | 4 | 4 | 4 | 0.5 | 4 |
| | | LR | 0.001 | 0.001 | 0.001 | 0.001 | 0.001 | 0.001 | 0.001 | 0.001 |
| | | Batch | 8 | 8 | 8 | 8 | 16 | 16 | 8 | 8 |
| | 24 | $\alpha$ | 4 | 4 | 4 | 0.5 | 0.5 | 0.5 | 4 | 0.5 |
| | | LR | 0.0005 | 0.001 | 0.0005 | 0.001 | 0.001 | 0.001 | 0.0005 | 0.001 |
| | | Batch | 8 | 32 | 16 | 16 | 32 | 16 | 32 | 16 |
| | 64 | $\alpha$ | 0.5 | 0.5 | 0.5 | 0.5 | 0.5 | 0.5 | 0.5 | 0.5 |
| | | LR | 0.001 | 0.0005 | 0.0005 | 0.001 | 0.001 | 0.0005 | 0.001 | 0.0005 |
| | | Batch | 32 | 8 | 32 | 16 | 32 | 8 | 8 | 8 |
| RoBERTa$_{Large}$ | 16 | $\alpha$ | 0.5 | 0.5 | 0.5 | 0.5 | 0.5 | 0.5 | 4 | 0.5 |
| | | LR | 0.001 | 0.001 | 0.0005 | 0.001 | 0.001 | 0.0005 | 0.001 | 0.001 |
| | | Batch | 8 | 32 | 32 | 32 | 16 | 8 | 16 | 32 |
| | 32 | $\alpha$ | 4 | 0.5 | 0.5 | 0.5 | 0.5 | 0.5 | 4 | 0.5 |
| | | LR | 0.0005 | 0.001 | 0.001 | 0.001 | 0.0005 | 0.0005 | 0.0005 | 0.001 |
| | | Batch | 32 | 16 | 32 | 32 | 8 | 32 | 32 | 16 |

Table 5: **Hyper-parameters for RoBERTa for MetaTT-4D.** We list here the hyper-parameters that can be used to replicate the results for MetaTT-4D reported in Table 2.

**Hyper-parameter search grid.** We also list the hyper-parameter grids we used to search for the set of hyper-parameters we reported for fine-tuning RoBERTa using MetaTT-4D and MetaTT-5D on GLUE benchmark datasets, in Table 5 and Table 6, in Table 7. Across both models and methods, we use 0.0 as weight decay, warmup ratio of 0.06, and set the sequence length at 256.

| Model | Rank | Params | CoLA | MNLI | MRPC | QNLI | QQP | RTE | SST2 | STS-B |
|---|---|---|---|---|---|---|---|---|---|---|
| RoBERTa$_{\text{Base}}$ | 16 | $\alpha$ | 0.5 | 0.5 | 0.5 | 0.5 | 0.5 | 0.5 | 0.5 | 0.5 |
| | | LR | 0.0005 | 0.001 | 0.001 | 0.001 | 0.001 | 0.001 | 0.0005 | 0.001 |
| | | Batch | 32 | 16 | 8 | 8 | 8 | 8 | 16 | 8 |
| | 64 | $\alpha$ | 0.5 | 0.5 | 0.5 | 0.5 | 0.5 | 0.5 | 0.5 | 0.5 |
| | | LR | 0.0005 | 0.0005 | 0.001 | 0.001 | 0.0005 | 0.0005 | 0.001 | 0.001 |
| | | Batch | 32 | 8 | 16 | 16 | 16 | 32 | 16 | 8 |
| RoBERTa$_{\text{Large}}$ | 32 | $\alpha$ | 0.5 | 0.5 | 0.5 | 0.5 | 0.5 | 0.5 | 0.5 | 0.5 |
| | | LR | 0.001 | 0.001 | 0.0005 | 0.001 | 0.001 | 0.0005 | 0.0005 | 0.001 |
| | | Batch | 32 | 8 | 32 | 32 | 16 | 8 | 8 | 16 |
| | 64 | $\alpha$ | 0.5 | 0.5 | 0.5 | 0.5 | 0.5 | 0.5 | 0.5 | 0.5 |
| | | LR | 0.0005 | 0.001 | 0.0005 | 0.0005 | 0.0005 | 0.0005 | 0.0005 | 0.0005 |
| | | Batch | 16 | 32 | 16 | 16 | 16 | 16 | 16 | 8 |

Table 6: **Hyper-parameters for RoBERTa for MetaTT-5D.** We list here the hyper-parameters that can be used to replicate the results for MetaTT-5D reported in Table 2.

| Hyper-parameter | MetaTT-4D Values | MetaTT-5D Values |
|---|---|---|
| Rank ($r$) | $4, 8, 16, 24, 32, 48, 64$ | Base $-\{16, 24, 32, 48, 64\}$. Large $-\{32, 64, 96\}$ |
| Alpha ($\alpha$) | $0.5, 4$ | $0.5, 4$ |
| Learning Rate ($\eta$) | $1 \times 10^{-3}, 5 \times 10^{-4}$ | $1 \times 10^{-3}, 5 \times 10^{-4}$ |
| Batch Size | $8, 16, 32$ | $8, 16, 32$ |

Table 7: Hyper-parameter grid used for RoBERTa$_{\text{base}}$ and RoBERTa$_{\text{large}}$ on GLUE benchmark datasets to fine-tune with MetaTT-4D and MetaTT-5D PEFT adapters.

**Hyper-parameter search for Llama.** Since fine-tuning on Llama models is computationally more demanding, we restrict the search of hyper-parameters over coarser grids in conjunction with some heuristics. Precisely, we perform a heuristic search over the grid spanned by the TT-ranks $r \in \{8, 16, 32, 64, 128, 256\}$, alpha values $\alpha \in \{1.0, 2.0, 3.0\}$, learning rates $\eta \in \{1e-4, 2e-4, 5e-4\}$, and over two epochs. All Llama results were obtained by initializing the two middle cores as Gaussians with std$= 0.2$ and mean 0, and the right core being set to identity (the left core being set to zero). For MetaTT and other baselines we use AdamW as optimizer along with a linear scheduler, a warmup ratio of 0.06, and effective batch size of 32.

**Hyper-parameters for MTL.** We used a fixed learning rate of $5e - 4$, and a weight decay schedule of $0.0$ for LoRA and the variants of MetaTT.

### D.5 BASELINES

Several of the baselines reported in Table 2 had extensively reported the set of hyper-parameters used to benchmark against LoRA: VeRA (Kopiczko et al., 2024, Tables 8, 9), LoRETTA (Yang et al., 2024, Tables 12, 13), LoRTA (Hounie et al., 2024, Tables 11, 12), LoTR (Bershatsky et al., 2024, §D), MTL-LoRA and MoE-LoRA (Yang et al., 2025, Table 7)). However, except for LoTR and LoRTA, all other methods report accuracy after fine-tuning the weights of both the classifier head and the shared parameters. However, allowing the whole classifier head to be trainable significantly blows up the total number of trainable parameters (e.g., adds about $400K$ parameters for RoBERTa$_{\text{base}}$ in case of VeRA with sequence length of $1024$), effectively hiding the sole impact of the shareable hyper-parameters. As such we re-run the benchmarking by freezing the classifier heads. We believe that this is necessary for a fair comparison. We also report the new set of hyper-parameters for replicating these results in the following subsections.

### D.5.1 LORA HU ET AL. (2021)

The reported results for fine-tuning Llama-2 models with LoRA were obtained through hyper-parameter tuning over a grid with ranks $r \in 8, 16, 32, 64$, alpha $\alpha = 2r$, and learning rates $\eta \in$

$1e$-$1, 1e$-$2, 1e$-$5$. The values presented correspond to the best performance achieved on this grid search over 2 epochs.

For RoBERTa, we directly report the results from Bershatsky et al. (2024).

### D.5.2 VERA KOPICZKO ET AL. (2024)

For each of the GLUE benchmark datasets we use weight decay $0.0$ and warmup ratio $0.06$. Furthermore, to be consistent with our other experiments, for RoBERTa$_{\text{base}}$ and RoBERTa$_{\text{large}}$, we use a sequence length of 256. We tried different batch sizes from the set $\{4, 8, 16, 32\}$ out of which 32 consistently worked well across all of the datasets. The best performing learning rates for both the models and corresponding datasets are reported in Table 8. To find these learning rates, we searched in the range $[0.0001, 0.1]$ across all datasets. Finally, consistent with the experiments in Kopiczko et al. (2024), we set VeRA rank for RoBERTa$_{\text{base}}$ as 1024, and for RoBERTa$_{\text{large}}$ as 256.

| Model | CoLA | MNLI | MRPC | QNLI | QQP | RTE | SST2 | STS-B |
|---|---|---|---|---|---|---|---|---|
| RoBERTa$_{\text{base}}$ | 0.005 | 0.0008 | 0.01 | 0.015 | 0.025 | 0.004 | 0.01 | 0.003 |
| RoBERTa$_{\text{large}}$ | 0.009 | 0.01 | 0.004 | 0.006 | 0.01 | 0.005 | 0.01 | 0.003 |

Table 8: **Learning rates for VeRA experiments.** Since we only train the attention layers and keep the classifier weights frozen, we report the learning rates to fine-tune RoBERTa$_{\text{base}}$ and RoBERTa$_{\text{large}}$ on the GLUE benchmark datasets.

For Llama experiments we perform a coarser grid search given runs are substantially more expensive. For each of the two models we searched over the grid formed by ranks $r \in \{256, 1024\}$ and learning rates $\eta \in \{2e-4, 5e-4\}$. We picked the best out of these parameters over the span of two epochs.

Other than rank and learning rate, for both RoBERTa and Llama experiments we use the default set of hyper-parameters from HuggingFace's implementation of VeRA.

### D.5.3 LORETTA YANG ET AL. (2024)

Similar to the original paper, for LoRETTA$_{adp}$ we tried on two batch sizes $16, 32$ and found 32 to work best in all tasks of the GLUE suite. The bottleneck dimension was set at 64. Adapter dropout was set to 0, and the scaling parameter ($\alpha$) was set at 1.0. Weight decay was set to 0.01 and sequence length was set at 256 for both the methods. Furthermore, among $2, 5, 10, 20$ tensor ranks, 5 had the right balance of parameters and performance across GLUE tasks for both the methods. The learning rates used across the tasks, method, and the models are reported in Table 9. Each of the dataset was trained on 20 epochs, except MNLI and QQP which were trained for 10 epochs.

| LoRETTA$_{adp}$ | | | | | | | | |
|---|---|---|---|---|---|---|---|---|
| Model | CoLA | MNLI | MRPC | QNLI | QQP | RTE | SST2 | STS-B |
| RoBERTa$_{\text{Base}}$ | $1e-3$ | $4e-4$ | $8e-4$ | $2e-3$ | $3e-3$ | $2e-3$ | $3e-4$ | –(–) |

| LoRETTA$_{rep}$ | | | | | | | | |
|---|---|---|---|---|---|---|---|---|
| Model | CoLA | MNLI | MRPC | QNLI | QQP | RTE | SST2 | STS-B |
| RoBERTa$_{\text{Base}}$ | $5e-4$ | $1e-4$ | $7e-4$ | $1e-3$ | –(–) | $4e-4$ | $3e-4$ | –(–) |

Table 9: **Learning rates for LoRETTA$_{\text{adp}}$ experiments.** Since we only train the attention layers and keep the classifier weights frozen, we report the learning rates to fine-tune RoBERTa$_{\text{Base}}$ and RoBERTa$_{\text{Large}}$ on the GLUE benchmark datasets.

### D.5.4 LOTR BERSHATSKY ET AL. (2024)

For RoBERTa experiments we directly quote the results of LoTR from the original work. For Llama models we search exhaustively over the grid spanned by the learning rates $\eta \in \{2e-4, 5e-4\}$ and ranks $r \in \{16, 64\}$. The choice of scaling factor $\alpha$ is set to 2.0 and we initialize the cores analogously to MetaTT: middle core drawn from a Gaussian with std$= 0.2$ and mean 0, and right core (output dimension leg) set to identity. We report the best accuracy on this grid over two epochs.

### D.5.5 LoRTA Hounie et al. (2024)

We chose to rerun GLUE experiments for LoRTA since in (Hounie et al., 2024, Table 2) only the best results are reported. For completeness, we restate the hyperparamter grids used to report results in Hounie et al. (2024) for GLUE baselines in Table 10. We train the models on the 3 seeds mentioned at the beginning of this section. For COLA, MRPC, STS-B, and RTE tasks we fine-tune RoBERTa based models for 20 epochs, and for MNLI, SST2, QNLI, and QQP tasks we fine-tune RoBERTa based models for 10 epochs. Similar to other experiments we set sequence length at 256. Weight decay is set at $0.0$ for fine-tuning both models. The learning rate grid is $[5e-4, 1e-3, 2e-3, 3e-3, 4e-3, 5e-3, 6e-3, 7e-3, 1e-2, 1.5e-2, 2e-2]$ for RoBERTa$_{base}$, and $[1e-3, 5e-3, 7e-3, 8e-3, 9e-3, 1e-2, 2e-2]$ for RoBERTa$_{large}$. The final learning rates used are reported in Table 11.

| Hyper-parameter | RoBERTa$_{base}$ | RoBERTa$_{large}$ |
|:---:|:---:|:---:|
| $\alpha$ | $[0.5, 1.0, 2.0, 8.0]$ | $[0.5, 1.0, 2.0, 8.0]$ |
| Scheduler | Linear | Linear |
| Optimizer | AdamW | AdamW |
| Batch size | $[32, 64]$ | $[32, 64]$ |
| Warmup ratio | 0.06 | 0.06 |

Table 10: **Hyper-parameter configurations for RoBERTa$_{base}$ and RoBERTa$_{large}$ for LoRTA.** We restate and update some of the hyper-parameters from Hounie et al. (2024) for completeness.

| Model | COLA | MNLI | MRPC | QNLI | QQP | RTE | SST2 | STS-B |
|:---:|:---:|:---:|:---:|:---:|:---:|:---:|:---:|:---:|
| RoBERTa$_{base}$ | $1e-2$ | $1e-2$ | $1e-2$ | $1e-2$ | $1.5e-2$ | $1e-2$ | $4e-3$ | $4e-3$ |
| RoBERTa$_{large}$ | $1e-2$ | $1e-2$ | $1e-2$ | $1e-2$ | $8e-3$ | $2e-2$ | $1e-2$ | $2e-2$ |

Table 11: **Final learning rates used for fine-tuning RoBERTa$_{base}$ and RoBERTa$_{large}$ using LoRTA.** We state the final learning rates used to fine-tune RoBERTa based pre-trained models on the GLUE tasks.

### D.5.6 MoE-LoRA Liu et al. (2024) and MTL-LoRA Yang et al. (2025)

The codebase for MTL-LoRA[1] had implementation only for Llama based models. As such we adapted this codebase for experimenting MTL on RoBERTa based models for fine-tuning on some of the GLUE tasks. For both MoE-LoRA and MTL-LoRA, and both RoBERTa based models, we train for 20 epochs. The number of experts were set to 4 for MoE-LoRA (used a grid of $[4, 8]$, 4 worked reliably well), and rank $r$ of LoRA was set to 4 (grid used was $[4, 8]$, 4 gave the best balance between number of trainable parameters and performance) for both methods. We set a batch size of 32 across both methods and the learning rate grid used was $[1e-4, 2e-4, 3e-4, 5e-4, 7e-4, 9e-4, 1e-3]$ for both the methods. The final learning rates used are reported in Table 12.

| Model | MoE-LoRA | MTL-LoRA |
|:---:|:---:|:---:|
| RoBERTa$_{base}$ | $7e-4$ | $5e-4$ |
| RoBERTa$_{large}$ | $9e-4$ | $5e-4$ |

Table 12: **Final learning rates for MTL experiments using methods from Liu et al. (2024); Yang et al. (2025).** In this table we report the final learning rates used to generate the baseline results in Table 3 using MoE-LoRA and MTL-LoRA.

## E    MetaTT Adapter Implementation

In this section, we give an example of one of the MetaTT adapters (MetaTT-4D) using pseudo-code written in python.

---

[1] https://github.com/pUmpKin-Co/MTL-LoRA

**Configuration.** We start by stating the configuration class of MetaTT-4D in Algorithm 2. The different inputs to the class function are – `rank`: a $1 \times 3$ array (e.g., $[8, 8, 8]$) corresponding to the different ranks of the tensor-train, `alpha`: scaling factor for the adapter, `target_modules`: type of matrices to be fine-tuned using the algorithm, and `use_bias`: flag to choose whether to add bias as a parameter.

---

**Algorithm 2** MetaTT-4D configuration class file

---

```python
class MetaTT4DConfig:
    def __init__(self, ranks, alpha=1.0, target_modules=["query", "key", "value"],
 use_bias=False):
        self.ranks = ranks
        self.alpha = alpha   # scaling factor for the adapter
        self.target_modules = target_modules
        self.use_bias = use_bias
```

---

**Adapter.** We then list an example of an adapter for MetaTT-4D in Algorithm 3. In the attached pseudo-code, the $\mathcal{G}_1$ core is initialized to zero, the $\mathcal{G}_2$, $\mathcal{G}_3$, $\mathcal{G}_4$ cores are initialized as the identity matrix. One of the cores is generally set to **0**-tensor so that output of the corresponding adapter is zero at the beginning of the training similar to Hu et al. (2021).

---

**Algorithm 3** MetaTT-4D adapter

---

```python
class MetaTT4DAdapter(nn.Module):
    def __init__(self, hidden_dim: int, num_layers: int, tt_config: MetaTT4DConfig):
        super().__init__()
        self.hidden_dim = hidden_dim
        self.num_layers = num_layers
        self.tt_config = tt_config
        self.num_projs = len(tt_config.target_modules)

        # initialize the tensor cores
        self.G1 = Parameter(torch.empty(self.hidden_dim, \
            tt_config.ranks[0]), requires_grad=True)

        self.G2 = ParameterList[nn.init.eye_(Parameter(torch.zeros\
            (self.tt_config.ranks[0], self.tt_config.ranks[1]),\
            requires_grad=True)) for _ in range(self.num_layers)]

        self.G3 = ParameterList[nn.init.eye_(Parameter(torch.empty\
            (self.tt_config.ranks[1], self.tt_config.ranks[2]),\
            requires_grad=True)) for _ in range(self.num_projs)]

        self.G4 = Parameter(torch.empty(tt_config.ranks[2],\
            self.hidden_dim), requires_grad=True)

        nn.init.zeros_(self.G1)
        nn.init.eye_(self.G4)
```

---

**Linear adapter and forward function.** Algorithm 3 can be used to create a linear adapter and the corresponding forward function as shown in Algorithm 4. The inputs to this adapter are – `original_layer`: the layers of the pre-trained model, `tt_config`: the corresponding initialized MetaTT configuration class, `M1`, `M2`, `M3`, `M4`: the tensor core slices along each of the four dimensions (corresponding to matrices). During the forward pass, the input batch is first passed through the original layer that is frozen. This batch is also multiplied by `M1` trough `M4`, and scaled by $\alpha$. Note that this order of matrix multiplication will be optimal if the rank is smaller than the batch size. The outputs of the original and the adapter are then added and returned.

**PEFT model.** Finally, the pseudo-code for developing our model with PEFT philosophy so that it can be used as a drop-in within any model quickly is given in Algorithm 5. The inputs to this algorithm is just the pre-trained model and the configuration class for MetaTT. Algorithm 5 uses the cores initialized in Algorithm 3 to set up the trainable layer using Algorithm 4. Finally, these

**Algorithm 4** MetaTT-4D linear adapter

```
class MetaTT4DLinearAdapter(nn.Module):
    def __init__(self, original_layer: nn.Module, tt_config: MetaTT4DConfig, M1, M2,
M3, M4):
        super().__init__()
        # set requires_grad to False for original weights
        self.original_layer = original_layer.\
                        requires_grad_(False)
        self.tt_config = tt_config
        self.M1 = M1
        self.M2 = M2
        self.M3 = M3
        self.M4 = M4

    def forward(self, X: torch.tensor) -> torch.Tensor:
        original_output = self.original_layer(X)
        return original_output + self.tt_config.alpha * (((X @ self.M1) @ self.M_2) @
self.M3) @ self.M4
```

layers are set-up such that one can dynamically update them during runtime. Once all the layers are initialized correctly, the model is assigned to the corresponding training device and returned.

**Algorithm 5** MetaTT-4D PEFT model for RoBERTa

```
def get_meta_tt_4d_model(model, config):
    # set pre-trained model weights to be non-trainable
    for param in model.parameters():
        param.requires_grad = False
    # grab device ID from model
    device = model.device

    num_layers = model.config.num_hidden_layers
    hidden_dim = model.config.hidden_size

    # initialize the MetaTT adapter
    meta_tt_adapter = MetaTT4DAdapter(hidden_dim, num_layers,\
                                      config)

    # go through each layer and corresponding projection
    # matrices of the roberta model
    for layer_idx, layer in \
            enumerate(model.roberta.encoder.layer):
        for proj_idx, proj_matrix in \
            enumerate(config.target_modules):
            if proj_matrix in ("query", "key", "value"):
                original_layer = layer.attention.self
                original_matrix = getattr(original_layer,\
                                          proj_matrix)
            elif proj_matrix == "dense":
                original_layer = layer.attention.output
                original_matrix = getattr(original_layer,\
                                          proj_matrix)
            else:
                raise ValueError(f\
                    "Unexpected proj_matrix value: {proj_matrix}")

            # set-up the MetaTT layer
            meta_tt_layer = MetaTT4DLinearAdapter(\
                            original_matrix, config,
                            meta_tt_adapter.G1,
                            meta_tt_adapter.G2[layer_idx],
                            meta_tt_adapter.G3[proj_idx],
                            meta_tt_adapter.G4)

            setattr(original_layer, proj_matrix, meta_tt_layer)

    return model.to(device)
```

## F    LIMITATIONS AND BROADER IMPACTS

**Limitations.**    MetaTT is sensitive to parameter initialization. We have observed that for certain choice of hyper-parameters MetaTT would fail to train. This sensitivity to initialization is more prevalent in the 5D version than in the 4D. Finding better initialization heuristics would improve the robustness of MetaTT. We also observe that when compared to MetaTT-4D, MetaTT-5D is more sensitive to worsening performance during training when optimized using standard SGD algorithms.

**Broader Impacts.**    The development of a reparameterization adapter using tensor trains and DMRG-inspired techniques offers significant potential in advancing compressed adapter fine-tuning as well as model training. By leveraging these methods, models can be compressed as they are being trained, significantly reducing the final parameter count while maintaining high accuracy. This leads to ultra-compressed models while training, and compressed adapters while fine-tuning. The ability to compress models as they are training ensures that the compression does not compromise the model's performance, as there are opportunities for correction during the training process itself.

Further, our work opens up new possibilities for deploying advanced scalable models in resource-constrained settings – where our techniques could allow for maintaining a high accuracy during training as opposed to approaches where a model is compressed post-training.

## G    REBUTTAL APPENDIX: ADDITIONAL EXPERIMENTS ON MTL

We add additional experiments for MTL in this section (and will collate and combine with Section 3.2 and Appendix B for the final release).

We fine-tune RoBERTa$_{base}$ and RoBERTa$_{large}$ jointly on CoLA, MRPC, RTE, and QNLI (MTL for 4 tasks) datasets from the GLUE benchmark datasets. In Table 13 we show the results of our experiments on MTL for 4 tasks. We observe similar patterns when compared to the results for MTL 3 tasks (fine-tune RoBERTa$_{base}$ and RoBERTa$_{large}$ jointly on CoLA, MRPC, and RTE). LoRA again works remarkably well across different datasets and pre-trained models. MoE-LoRA outperforms methods other than LoRA across different tasks for RoBERTa$_{base}$, and MetaTT-(4+1)D outperforms methods other than LoRA across all tasks when RoBERTa$_{large}$ is fine-tuned. However, we observe that on average MetaTT-4D marginally outperforms MetaTT-(4+1)D when RoBERTa$_{base}$ is fine-tuned and is about $1\%$ worse when RoBERTa$_{large}$ is fine-tuned. We note that fine-tuning both models over these 4 tasks only incurs less than 1% extra parameters compared to the 3 tasks considered in the main text.

| Model | Method | Param $\times 10^3$ | Rank | Metric (%) | | | | |
|---|---|---|---|---|---|---|---|---|
| | | | | CoLA | MRPC | RTE | QNLI | Avg |
| RoBERTa$_{base}$ | LoRA | 295 | 8 | **58.6(9)** | 87.1(5) | 75(1) | **88.0(5)** | **77.3(3)** |
| | MTL-LoRA Yang et al. (2025) | 296 | 4 | 51.5(1) | 87(1) | **76.4(2)** | 87.6(5) | 75.6(5) |
| | MoE-LoRA Liu et al. (2024) | 309 | 8 | 54.5(3) | **87.4(8)** | **75.6(9)** | **88.2(3)** | **76.4(6)** |
| | MetaTT-4D | 13.2 | 8 | 53.8(5) | 84.8(3) | 75(2) | 85.9(8) | 74.8(7) |
| | MetaTT-(4+1)D | 13.5 | 8 | 54(1) | **87.3(9)** | 70(2) | 85.8(4) | 74.5(4) |
| RoBERTa$_{large}$ | LoRA | 786 | 8 | **65(1)** | **89(1)** | 80.8(8) | **92.9(4)** | **81.7(5)** |
| | MTL-LoRA Yang et al. (2025) | 789 | 4 | 60(1) | 88.6(8) | 82(1) | 90.2(3) | 80.4(9) |
| | MoE-LoRA Liu et al. (2024) | 814 | 8 | 63(2) | **89(2)** | **83(2)** | 90.2(1) | 81(1) |
| | MetaTT-4D | 18.0 | 8 | 59(1) | 88.2(5) | 82.4(4) | 90.3(5) | 80.0(5) |
| | MetaTT-(4+1)D | 18.3 | 8 | **63(1)** | 87.6(8) | **83.4(9)** | **91.1(2)** | **81.2(5)** |

Table 13: **Results of MTL with 4 tasks.** We observe that unlike results in Table 3 MetaTT-(4+1)D outperforms MetaTT-4D only when using RoBERTa$_{large}$. However, across both models, MetaTT-(4+1)D uses about only 300 more trainable parameters. For RoBERTa$_{base}$ MetaTT-(4+1)D performs within $1\%$ of MTL-LoRA while using $\approx 22$x less parameters, and for RoBERTa$_{large}$ MetaTT-(4+1)D outperforms both MTL-LoRA and MoE-LoRA on average, and is within $0.5\%$ of average accuracy of LoRA, while using $\approx 43$x less parameters. We show in bold the two best accuracies per task.

Similar to Appendix B we also plot the heatmaps of the gradients across each tensor in the TT Figure 5. We observe that across tensor slices which correspond to the specific tasks, MetaTT-(4+1) acquires large gradients in label 3 for CoLA (similar to what we saw in Appendix B). We also complement plots of the gradients with downstream task performance per epoch.

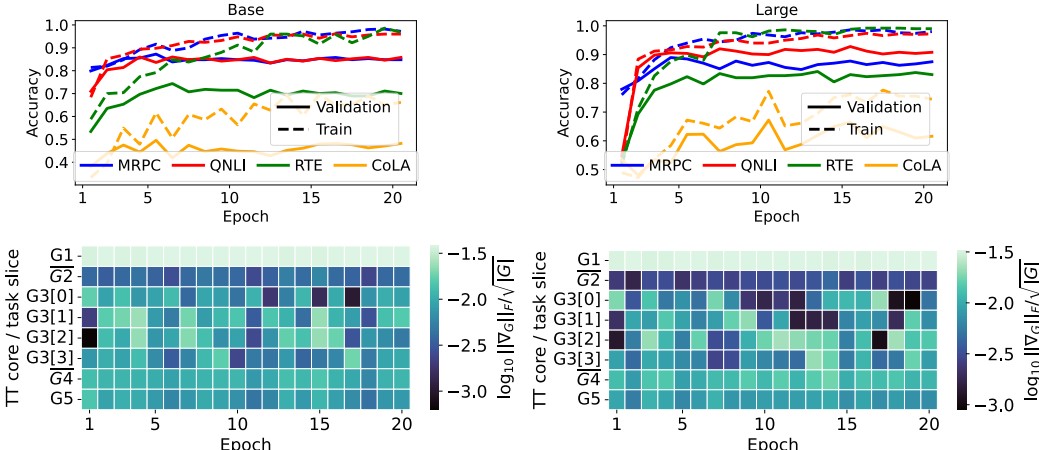

Figure 5: **Influence of task-dependent TT core in MTL.** (Left): (Top): accuracy of MetaTT-(4+1)D as a function of epochs for RoBERTa$_\text{Base}$ for a single training realization (in the case of CoLA we compute Matthew's correlation instead). (Bottom): Corresponding normalized gradients across all tensors as a function of epochs (see Appendix B). Task labels correspond to 0: MRPC, 1: QNLI, 2: RTE, 3: CoLA. (Right): Same as in left but for RoBERTa$_\text{Large}$ as pretrained model.

## H    REBUTTAL APPENDIX: STABILITY ANALYSIS

We compare stability of the performance of LoRA, VeRA, MetaTT4D, and MetaTT5D for fine-tuning RoBERTa$_\text{base}$ and RoBERTa$_\text{large}$ on the validation data of CoLA, MRPC and RTE. We study the stability of these adapters when ranks and learning rates are varied while keeping other hyperparameters constant.

### H.1    STABILITY DURING TRAINING USING THE BEST SET OF HYPERPARAMETERS

**Fixed hyper-parameters.**    For these experiments, we fix the random seeds to $\{1, 2, 3, 4, 5, 6, 7, 8\}$ across all algorithms and models. The hyper-parameters used for LoRA is reported in Table 14. Note, Bershatsky et al. (2024) which was used to report LoRA values in Table 2 only provided the range of hyper-parameter values swept, and not the final set of hyper-parameters, and so we ran our own search on a wider set of seeds. The hyper-parameters for other methods are consistent with the best set of hyper-parameters reported in Appendix D

| Model | Task | Rank | $\alpha$ | Learning Rate | Batch Size |
|---|---|---|---|---|---|
| | CoLA | 4 | 8.0 | $5e-4$ | 16 |
| RoBERTa$_\text{base}$ | MRPC | 32 | 64.0 | $2e-4$ | 8 |
| | RTE | 8 | 16.0 | $2e-4$ | 16 |
| | CoLA | 64 | 128.0 | $2e-4$ | 16 |
| RoBERTa$_\text{large}$ | MRPC | 32 | 64.0 | $2e-4$ | 8 |
| | RTE | 8 | 16.0 | $2e-4$ | 16 |

Table 14: Best hyperparameters for each model and task combination for LoRA.

**Stability score.**    To quantify stability of each adapters on the best corresponding set of hyperparameters, we report margin of error (also known as the half-width of the $95\%$ confidence interval) Krishnamoorthy (2006) defined as

$$\text{Margin of error} \approx 1.96 \frac{\sigma}{\sqrt{n}}, \tag{7}$$

where 1.96 is the critical value expressed as a $z$-score and corresponds to $95\%$ confidence level, assuming that the data is normally distributed. Note, the margin of error reported here is at most a factor of 2 worse than the standard error.

**Observations.** In Figure 6 we plot the evaluation accuracy as we increase epochs during training. We observe that all methods demonstrate similar variance, across seeds and epochs. To measure the stability of the performance of the final model, we measure corresponding stability scores and report them in Table 15. We observe that between models, tasks and adapters, the half width of the 95% confidence interval does not vary by much. As a result, we can conclude that for the set of hyper-parameters that yield the highest evaluation accuracies across methods, the stability during training across adapter variants remains relatively similar.

| Model | Task | LoRA | VeRA | MetaTT-4D | MetaTT-5D |
|-------|------|------|------|-----------|-----------|
| RoBERTa$_{base}$ | CoLA | **0.009** | 0.012 | 0.012 | **0.011** |
| | MRPC | **0.006** | **0.006** | **0.006** | **0.006** |
| | RTE | **0.007** | 0.016 | 0.011 | **0.010** |
| RoBERTa$_{large}$ | CoLA | **0.007** | 0.013 | 0.154 | **0.010** |
| | MRPC | **0.004** | 0.007 | **0.006** | 0.065 |
| | RTE | 0.011 | **0.008** | **0.010** | 0.093 |

Table 15: **Margin of error across seeds.** We report the half-width of the 95% confidence interval for finetuning RoBERTa$_{base}$ and RoBERTa$_{large}$ using LoRA, VeRA, MetaTT-4D, and MetaTT-5D adapters. The lowest two values in each row are shown in bold.

## H.2 STABILITY ACROSS LEARNING-RATES

We also plot the performance of each of the adapters when the learning rate is varied while other hyper-parameters are kept constant in Figure 7 and Figure 8 (we choose the best reported hyper-parameters for these sweeps in Appendix D). We observe that when finetuning RoBERTa$_{base}$ and RoBERTa$_{large}$ on MRPC and RTE, the decay in performance after attaining the best rate is similar in LoRA and VeRA, and MetaTT-4D also performs similar to these adapters (approximately similar slope of the mean). For fine-tuning on CoLA, MetaTT-4D and LoRA behave similarly (with larger variance in case of MetaTT-4D), and MetaTT-5D and VeRA behave similarly. One must note that the $\alpha$ is varied for LoRA (as $2 \times \text{rank}$) and VeRA doesn't depend on $\alpha$, which makes this comparison harder for both variants of MetaTT. We observe similar trends across tasks for RoBERTa$_{large}$ for LoRA and MetaTT-4D. However, unlike RoBERTa$_{base}$, MetaTT-5D performs similar to LoRA and MetaTT-5D and not VeRA.

## H.3 STABILITY ACROSS RANKS

We also plot the performance of each of the adapters when the rank is varied while other hyper-parameters are kept constant in Figure 9 and Figure 10 (similar to Appendix H.2). Again note that the parameter $\alpha$ was chosen as $2 \times \text{rank}$ for LoRA, and for MetaTT variants were kept as fixed as reported in Appendix D. We first observe that across tasks and models sizes, LoRA and VeRA performs similarly across ranks. This is somewhat expected as we vary both rank and $\alpha$ (implicitly for VeRA and explicitly for LoRA). However, MetaTT-4D somewhat performs worse at higher ranks when both models are fine-tuned with CoLA and somewhat maintains performance on other tasks. However, MetaTT-5D's performance improves with ranks across models and ranks even with fixed $\alpha$.

## H.4 DIFFERENT INITIALIZATIONS OF LORA

Finally, we also plot the evaluation accuracy during training for several initializations of LoRA. Specifically, we compare the following initializations for LoRA – 1) Gaussian, 2) Pissa, and 3) OLoRA. We run 8 independent trials with the hyper-parameters in Table 17 and finetune RoBERTa$_{base}$ on MRPC and RTE tasks. We plot these results in Figure 11 We compare the validation accuracy across training epochs similar to Figure 4. We observe that across both MRPC and RTE, LoRA performs similarly when initialized with either Gaussian, Pissa Meng et al. (2024)(matrices initialized as singular vectors of the pre-trained weights), and OLoRA Büyükakyüz (2024) (base weights are translated with their QR decomposition). Although not as close the performance of these variants, we tested on different initializations of MetaTT-4D in Appendix D, and observed that the models

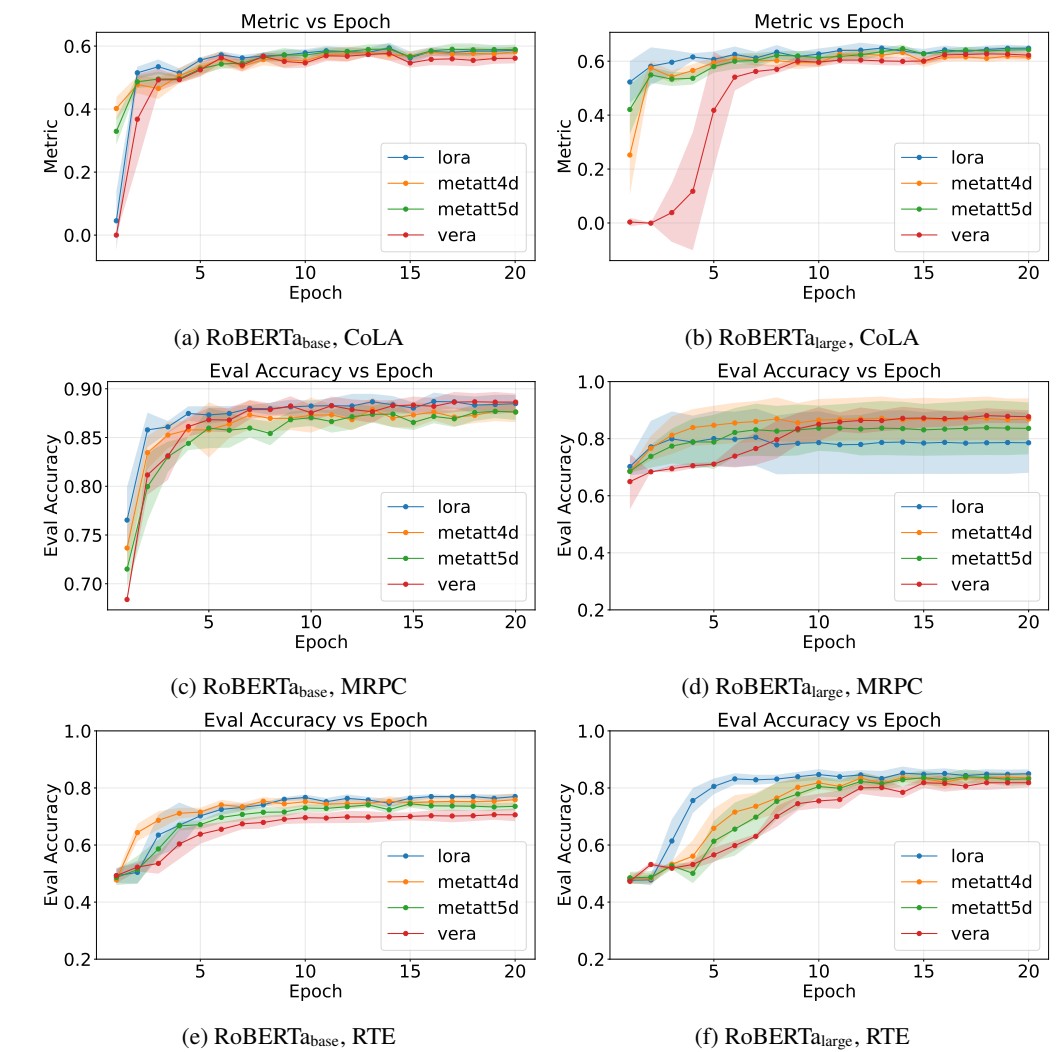

(a) RoBERTa$_{base}$, CoLA

(b) RoBERTa$_{large}$, CoLA

(c) RoBERTa$_{base}$, MRPC

(d) RoBERTa$_{large}$, MRPC

(e) RoBERTa$_{base}$, RTE

(f) RoBERTa$_{large}$, RTE

Figure 6: **Variance during training.** Here we plot the evaluation accuracy (Matthew's correlation coefficient for CoLA) as we train the respective model on specific finetuning tasks using LoRA, VeRA, MetaTT-4D, and MetaTT-5D adapters. We observe that on average the variance across LoRA, MetaTT-4D, and MetaTT-5D are approximately similar,except for RoBERTa$_{base}$ on CoLA where MetaTT-5D demonstrates significantly greater variance in training across seeds.

performed similarly across training epochs. However, tricks like the use of singular vectors of the pre-trained model to initialize adapter weights do not necessarily translate to the TT architecture.

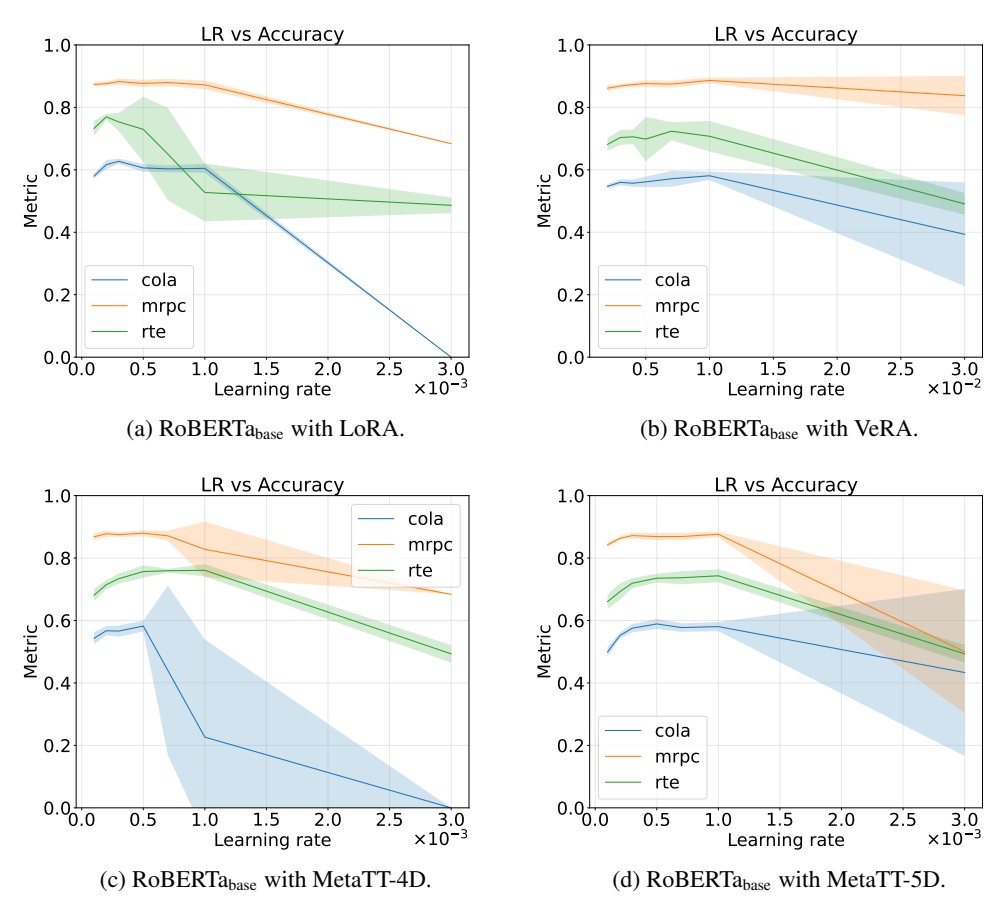

(a) RoBERTa$_{base}$ with LoRA.

(b) RoBERTa$_{base}$ with VeRA.

(c) RoBERTa$_{base}$ with MetaTT-4D.

(d) RoBERTa$_{base}$ with MetaTT-5D.

Figure 7: **Learning rate vs accuracy.** We plot final accuracy of RoBERTa$_{base}$ when trained with specific adapters on specific glue tasks. We observe that in general variants of MetaTT decays somewhat at a similar rate when compared to LoRA and VeRA. This is inspite of the fact that for LoRA and VeRA the parameter $\alpha$ are tied to the ranks and so varied across runs (or in case of VeRA already absorbed in the learning parameters), while $\alpha$ remains an independent parameter for variants of MetaTT and was treated asfixed across learning rates.

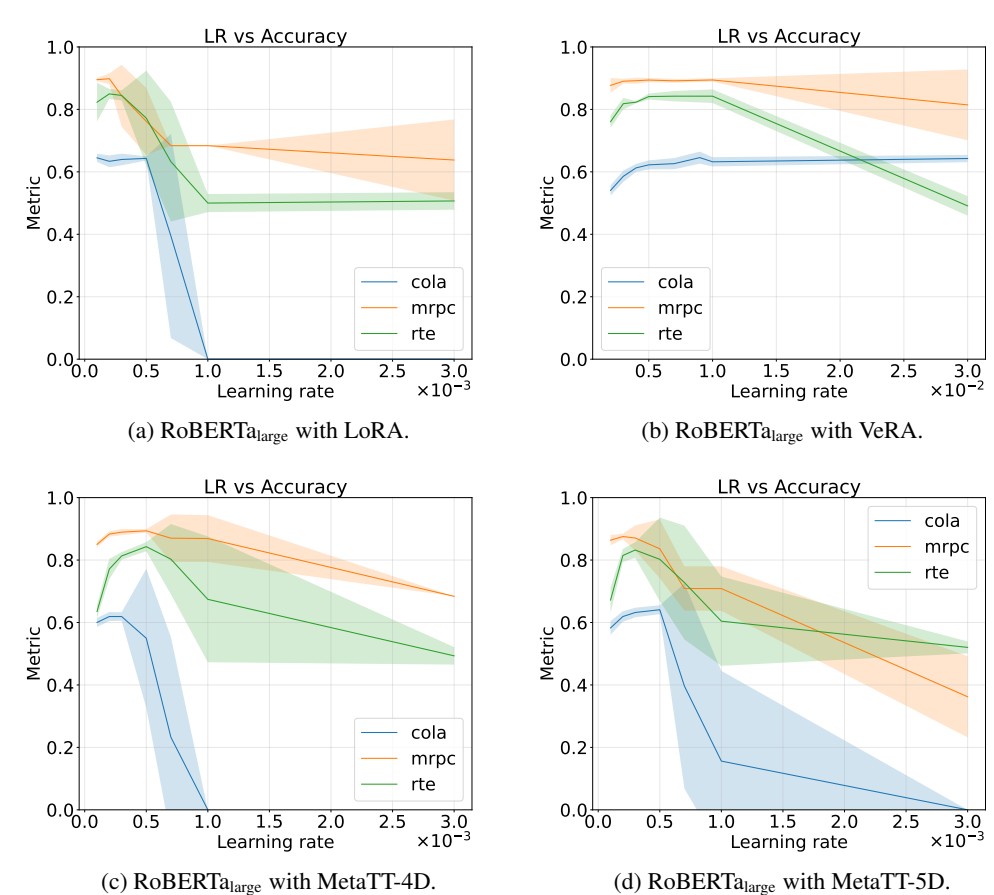

(a) RoBERTa$_{large}$ with LoRA.

(b) RoBERTa$_{large}$ with VeRA.

(c) RoBERTa$_{large}$ with MetaTT-4D.

(d) RoBERTa$_{large}$ with MetaTT-5D.

Figure 8: **Learning rate vs accuracy.** We plot final accuracy of RoBERTa$_{large}$ when trained with specific adapters on specific glue tasks. We observe that for MRPC the rate of performance decay for VeRA is better than other methods, and for RTE the rate of performance decay for MetaTT-5D is better than other methods.

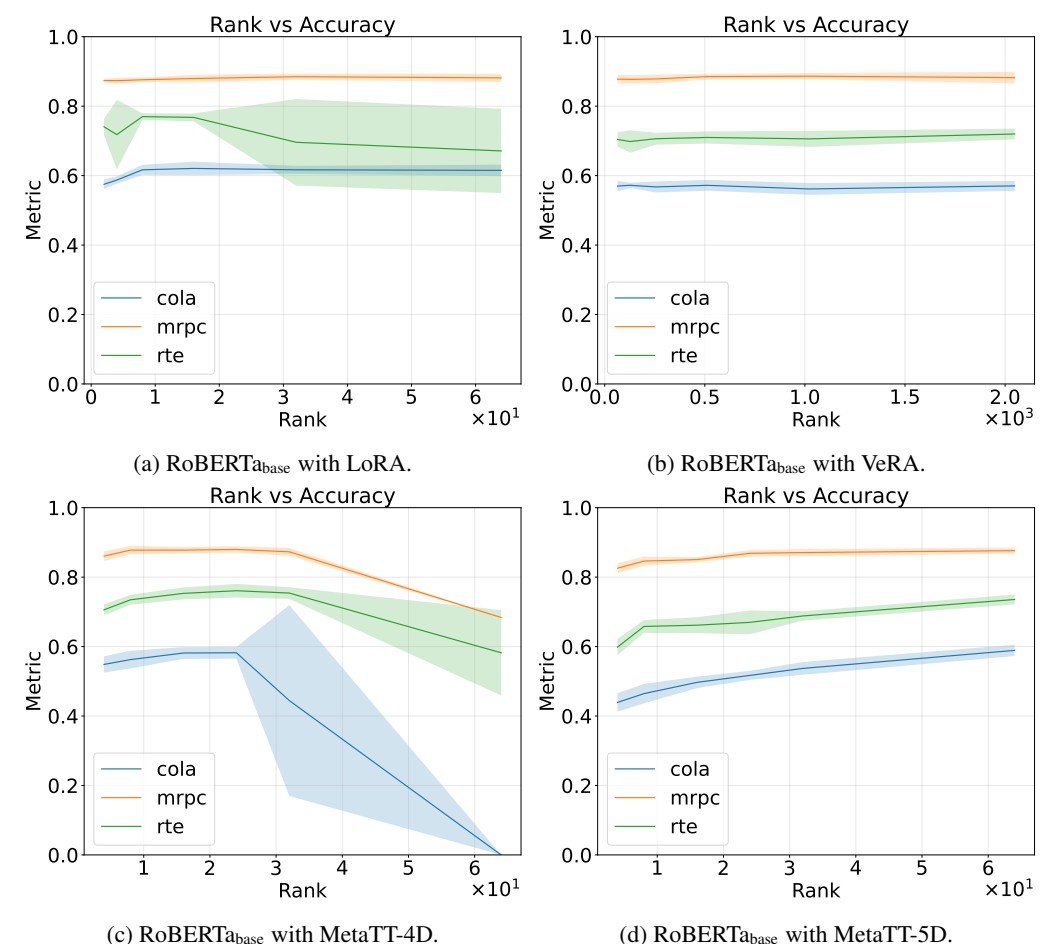

(a) RoBERTa$_{base}$ with LoRA.

(b) RoBERTa$_{base}$ with VeRA.

(c) RoBERTa$_{base}$ with MetaTT-4D.

(d) RoBERTa$_{base}$ with MetaTT-5D.

Figure 9: **Rank vs accuracy.** We plot final accuracy of RoBERTa$_{base}$ when trained with specific adapters on specific glue tasks when keeping other hyper-parameters fixed and varying ranks. We observe that both LoRA and VeRA maintains performance (hinting a little at model capacity). MetaTT-4D's performance gets somewhat worse at higher rank, hinting at the requirement to find the right pair of ranks and $\alpha$, while MetaTT-5D starts worse and keeps improving across ranks.

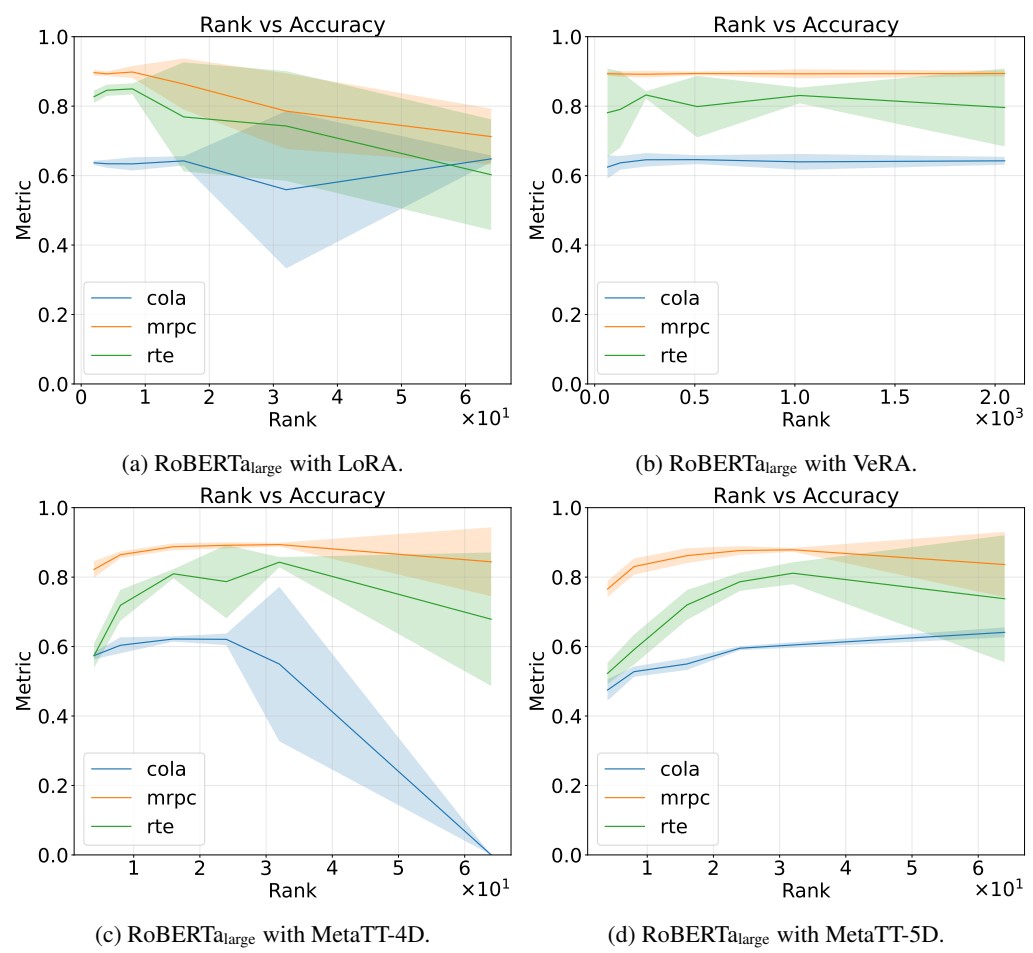

(a) RoBERTa_large with LoRA.

(b) RoBERTa_large with VeRA.

(c) RoBERTa_large with MetaTT-4D.

(d) RoBERTa_large with MetaTT-5D.

Figure 10: **Rank vs accuracy.** We plot final accuracy of RoBERTa_large when trained with specific adapters on specific glue tasks when keeping other hyper-parameters fixed and varying ranks. We observe that similar to Figure 9, LoRA and VeRA maintains performance across ranks. MetaTT-4D gets worse on CoLA while almost maintaining performance on MRPC and RTE, while MetaTT-5D starts worse but improves as we increase ranks.

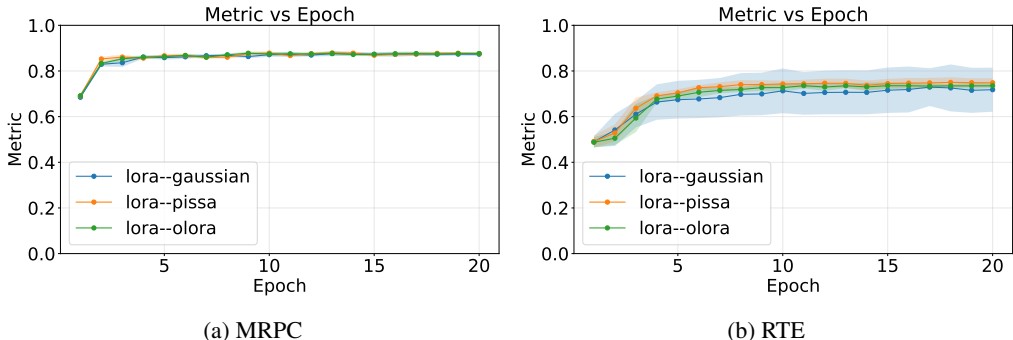

Figure 11: **LoRA using different initializations.** We plot the validation accuracy across training epochs using different initializations for LoRA on RoBERTa$_{\text{Base}}$. We observe that on average these initializations work similar to each other.

## I    REBUTTAL APPENDIX: ADALORA VS. METATT WITH DMRG-INSPIRED SWEEPS

### I.1    COMPARING ADALORA WITH LORA AND RELATING TO THE IMPROVEMENTS VIA DMRG

In this section we compare the performance of AdaLoRA Zhang et al. (2023) with LoRA. We want to specifically understand the capacity of a model trained on AdaLoRA for a fixed target rank to improve upon the performance of the LoRA with same rank. To establish fair comparison to LoRA we fix a target rank of 4 and report the results in Table 16. We report the mean and standard deviation of runs corresponding to seeds [33305628, 2025, 42] for RoBERTa$_{\text{base}}$ and [56346, 2025, 42] for RoBERTa$_{\text{large}}$. The best set of hyper-parameters found for AdaLoRA and LoRA for these experiments are reported in Table 17. In Figure 12 we also plot the validation accuracy during training RoBERTa$_{\text{base}}$ using both LoRA and AdaLoRA on three GLUE tasks. As with other experiments reported in our work, we again freeze the classifier and observe that AdaLoRA fails catastrophically for RTE. Moreover, in the cases where on average it outperforms LoRA, the variance of the resulting model is often higher. The fixed settings used for AdaLoRA were – 1) warmup steps was set at 200, number of steps for final finetuning was set at 1000, time interval between budget allocations was set at 10, hyperparameter for EMA sensitivity smoothing was 0.85 and for uncertainty quantification was 0.85 (used in the original paper), and total training steps was set at 2000. For $\alpha = 16$ and batch size 32, we searched for best learning rates in range $[1e-4, 1e-3]$ and report the corresponding best set of hyper-parameters.

| Model | Dataset | LoRA | AdaLoRA |
|---|---|---|---|
| RoBERTa$_{\text{base}}$ | CoLA | **60.8(5)** | 56.0(4) |
| | MRPC | 87(1) | **87.5(4)** |
| | RTE | **75(2)** | 52(2) |
| RoBERTa$_{\text{large}}$ | CoLA | 63.6(4) | **64.4(9)** |
| | MRPC | 89(1) | **90.3(4)** |
| | RTE | **85(2)** | 56(4) |

Table 16: **LoRA vs AdaLoRA.** We report the mean accuracy achieved when the target rank for LoRA and AdaLoRA is 4 while fine-tuning RoBERTa$_{\text{base}}$ and RoBERTa$_{\text{large}}$ in some GLUE tasks.

### I.2    COMPARISON OF ADALORA VS. DMRG

In Figure 13, we compare MetaTT-4D adapters (with and without DMRG-inspired sweeps) against LoRA and AdaLoRA adapters. Experiments are conducted on Commonsense15k, a downsampled version of Commonsense170k from Hu et al. (2023), using Llama-2-7b as the base model.

For MetaTT, we employ the following rank schedule $r(i) = r_f + (r_0 - r_f)\left[1 - (i/N)^\gamma\right]$, where $N = 5$, $i = 1, \ldots, 5$, $\gamma = 2$, initial rank $r_0 = 40$, and final rank $r_f = 20$. This schedule is motivated

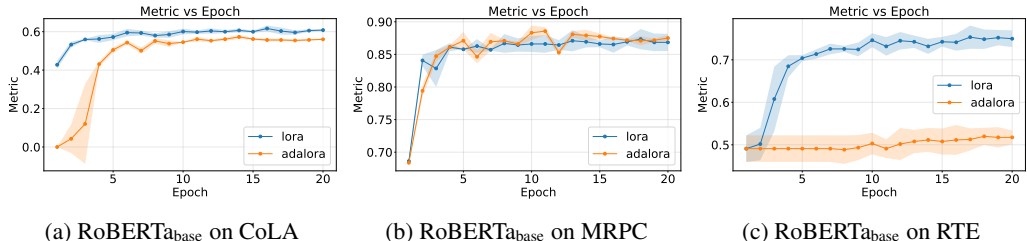

(a) RoBERTa$_{\text{base}}$ on CoLA  (b) RoBERTa$_{\text{base}}$ on MRPC  (c) RoBERTa$_{\text{base}}$ on RTE

Figure 12: **Comparison of LoRA and AdaLoRA during training.** We plot here the validation accuracy achieved using RoBERTa$_{\text{base}}$ on CoLA (Matthew's correlation coefficient), MRPC (evaluation accuracy), and RTE (evaluation accuracy) when trained using LoRA with rank 4 and AdaLoRA with initial rank 8 and final rank 4.

| Model | LoRA parameters | CoLA | MRPC | RTE | AdaLoRA parameters | CoLA | MRPC | RTE |
|---|---|---|---|---|---|---|---|---|
| | Rank | 4 | 4 | 4 | Init & target ranks | $[8, 4]$ | $[8, 4]$ | $[8, 4]$ |
| RoBERTa$_{\text{base}}$ | $\alpha$ | 8.0 | 8.0 | 8.0 | $\alpha$ | 16.0 | 16.0 | 16.0 |
| | Learning rate | $5e-4$ | $2e-4$ | $2e-4$ | Learning rate | $2e-4$ | $1e-3$ | $2e-4$ |
| | Batch | 16 | 8 | 16 | Batch | 8 | 8 | 8 |
| | Rank | 4 | 4 | 4 | Init & target ranks | $[8, 4]$ | $[8, 4]$ | $[8, 4]$ |
| RoBERTa$_{\text{large}}$ | $\alpha$ | 8.0 | 8.0 | 8.0 | $\alpha$ | 16.0 | 16.0 | 16.0 |
| | Learning rate | $2e-4$ | $2e-4$ | $2e-4$ | Learning rate | $2e-4$ | $4e-4$ | $2e-4$ |
| | Batch | 16 | 8 | 16 | Batch | 8 | 8 | 16 |

Table 17: **LoRA and AdaLoRA hyper-parameters used for fine-tuning.** Here we report the best set of hyper-parameters for LoRA and AdaLoRA found after performing hyper-parameter optimization when target rank for both methods is 4.

by empirical observations that larger rank reductions are more effective at later training steps, after the initial rapid learning phase. This approach closely resembles the cubic schedule used by AdaLoRA Zhang et al. (2023).

We implement AdaLoRA using its HuggingFace implementation with the following parameters: `target_r=8`, `init_r=16`, `tinit=84`, `tfinal=39`, `deltaT=84`, `beta1=0.85`, `beta2=0.85`, `orth_reg_weight=0.1`.

Our results show that MetaTT-4D with DMRG-inspired sweeps not only outperforms the counterpart trained simply via AdamW, for a target rank of $r = 20$, but also other variants with larger ranks, and more importantly LoRA and AdaLoRA adapters. We find that MetaTT-4D achieves accuracies comparable with LoRA with rank $r = 16$, but with $\approx 47$x fewer parameters.

In Table 18 we compare times taken for each of the runs on a single A100 GPU. We observe a slight overhead of running MetaTT with DMRG-inspired sweeps. This is mostly a result of a temporary increase in evaluation time immediately following each DMRG update. This artifact arises from PyTorch's need to recompile the computational graph and reinitialize CUDA kernels after the adapter tensor shapes are modified, rather than from the algorithmic complexity of the DMRG procedure itself. This motivates to apply fewer such DMRG steps (3-5 in our experiments).

| | LoRA ($r = 16$) | AdaLoRA ($r : 16 \to 8$) | MetaTT-4D ($r = 40$) | MetaTT-4D + DMRG ($r : 40 \to 20$) |
|---|---|---|---|---|
| Time (s) | 1725 | 1760 | 1680 | 1920 |

Table 18: **Training times for LoRA and MetaTT based adapters.** End-to-end training times for a single realization used in Fig. 13.

### I.3 COMPLEXITY ANALYSIS OF SVD-BASED RANK ADAPTATION: LORA VS. METATT

A key motivation behind the development of AdaLoRA is the computational expense associated with performing SVDs on all weight matrices to dynamically truncate LoRA ranks during training. This process quickly becomes prohibitive for large models, as the cost of SVD scales cubically with the hidden dimension. To mitigate this, AdaLoRA instead employs a pruning strategy, zeroing

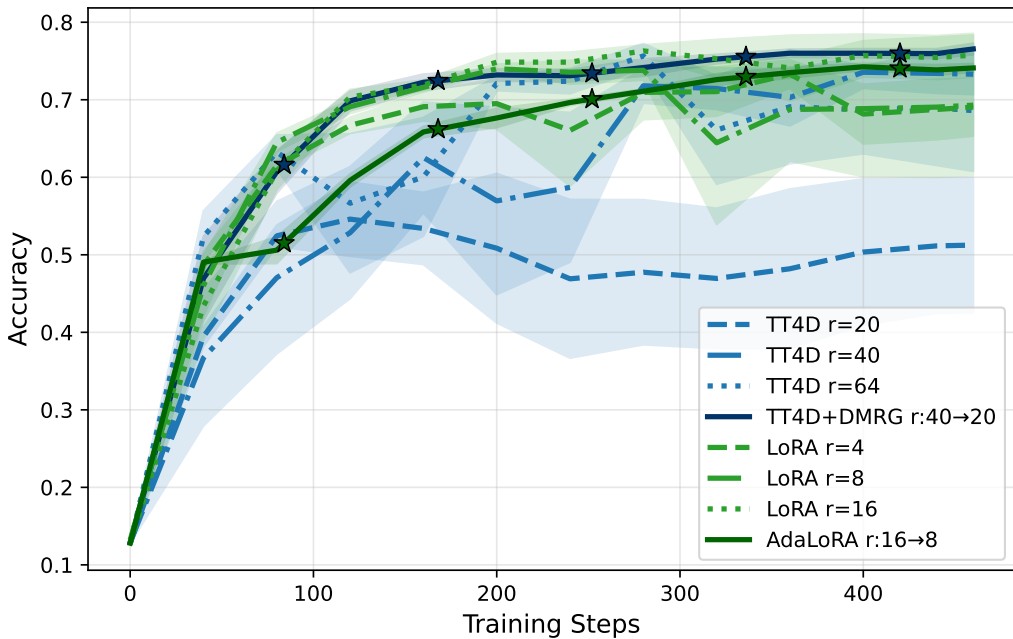

Figure 13: **Comparison of LoRA and AdaLoRA *vs.* MetaTT-4D and MetaTT-4D with DMRG-style sweeps.** Fine-tuning is performed on Commonsense15k (downsampled from Commonsense170k) with Llama-2-7b as the base model, for one epoch. Results show means and relative errors over 5 independent runs for each method. Star symbols indicate steps at which DMRG/AdaLoRA updates are applied. For MetaTT, $\alpha = 1.0$ (fixed); for LoRA, $\alpha = 2r$; for AdaLoRA, $\alpha = 32$ (twice the initial rank). Hyper-parameter tuning was performed over learning rates $[2e-4, 5e-4]$ for all adapters. See main text for details on the rank schedules used for MetaTT+DMRG and AdaLoRA.

out entries deemed irrelevant according to a score that serves as a proxy for singular values. In contrast, our DMRG-inspired algorithm for MetaTT adapters enables SVD-controlled truncations to be performed efficiently. The TT structure of MetaTT allows for global compression and facilitates rank adaptation via SVD sweeps over a much smaller set of tensor cores, rather than all individual weight matrices. This approach not only reduces computational overhead but also allows for dynamic reduction of matrix sizes during training, which is better exploited by GPUs compared to the sparse matrix operations required by AdaLoRA. To explicitly quantify the computational benefits of our rank-adaptive scheme over the alternative of performing SVDs on all weight matrices, we present a complexity analysis. The cost of performing a single series of SVDs for LoRA adapters is given by

$$O(LMD^3), \quad \text{(LoRA-SVD)} \tag{8}$$

where $L$ is the number of layers, $M$ is the number of matrices adapted per layer, and $D$ is the hidden dimension. In contrast, a single DMRG-style sweep (Algorithm 1) with initial TT-rank $r$ incurs a cost for MetaTT-4D of

$$O(2DLr\min(D, Lr)) + O(2LMr^2\min(Lr, Mr)) + O(2DMr\min(D, Mr)). \quad \text{(MetaTT-SVD)} \tag{9}$$

Assuming constant factors of O(1) in (8)–(9), the ratio of the LoRA-SVD cost to the MetaTT-SVD cost for the models considered in this work is summarized in Table 19 for various TT-ranks $r$

As shown in Table 19, the computational savings achieved by MetaTT are substantial, especially for moderate TT-ranks. For example, at $r = 16$, the MetaTT approach is over two orders of magnitude more efficient than LoRA-SVD for all models considered. This efficiency gain enables practical rank-adaptive training via SVD sweeps, which would otherwise be a computational overhead for large-scale models using conventional LoRA adapters.

| TT-rank $r$ | RoBERTa$_{Base}$ | RoBERTa$_{Large}$ | Llama-2-7b | Llama-2-13b |
|:---:|:---:|:---:|:---:|:---:|
| 16 | 186 | 169 | 2039 | 2553 |
| 64 | 11 | 16 | 127 | 159 |
| 256 | 2 | 3 | 16 | 20 |

Table 19: **Complexity of LoRA-SVD vs. MetaTT-SVD.** We show the ratio of LoRA-SVD to MetaTT-SVD complexity as the ratio of Eqs. (8) to (9) with an O(1) constant, for various TT-ranks $r$ and models.

## J   REBUTTAL APPENDIX: COMPARISON OF ACTUAL RUNTIMES ACROSS ADAPTERS

In Table 20 we compare the world clock runtimes of LoRA, VeRA, MetaTT-4D and MetaTT-5D for both forward pass and one gradient step in RoBERTa$_{base}$ and RoBERTa$_{large}$. For this we use a 1 A100 GPU node, which has 55 Intel Xeon Platinum CPUs, and 500 GBs RAM. We take a random sample of 5 batches and make forward and backward passes. We this 5 times and report the mean and standard deviation. For RoBERTa$_{base}$ and RoBERTa$_{large}$, we observe that LoRA and MetaTT-4D are the two of the fastest adapters in real world clock time. Note, the ranks chosen here are the ranks that were used to report results in Table 2.

| Model | Adapter | Rank | Batch | Forward pass (secs) | Backward pass (secs) |
|:---|:---|:---:|:---:|:---:|:---:|
| RoBERTa$_{base}$ | LoRA | 8 | 64 | **0.1859(1)** | **0.2014(1)** |
| | VeRA | 1024 | 64 | 0.2539(0) | 0.2718(1) |
| | MetaTT-4D | $24 \times 3$ | 64 | **0.1866(1)** | **0.2031(0)** |
| | MetaTT-5D | $64 \times 4$ | 64 | 0.1921(1) | 0.2146(1) |
| RoBERTa$_{large}$ | LoRA | 8 | 64 | **0.62(2)** | **0.6575(1)** |
| | VeRA | 256 | 64 | 0.6465(2) | 0.7035(3) |
| | MetaTT-4D | $32 \times 3$ | 64 | **0.6087(0)** | **0.6612(1)** |
| | MetaTT-5D | $64 \times 4$ | 64 | 0.63(1) | 0.6917(2) |

Table 20: **World clock runtime comparison of different PEFT adapters.** We report the average time required to make 5 batches pass through the model and corresponding adapter and then compute gradients, 5 independent trials. We observe that in general LoRA and MetaTT-4D are the fastest among the other methods reported here for their respective best performing ranks. We note that even for similar number of trainable parameters in VeRA, the matrix-vector-vector-matrix operation is significantly larger than the matrix-matrix operations in LoRA and variants of MetaTT, leading to gains in the forward and backward pass across batch and trials.

