# OpenReview forum: "MetaTT: A Global Tensor-Train Adapter for Parameter-Efficient Fine-Tuning"
_ICLR.cc/2026/Conference — Submitted to ICLR 2026_

### Official Review · Reviewer_6E7j · 2025-10-27

**Soundness:** 2
**Presentation:** 3
**Contribution:** 2
**Rating:** 4
**Confidence:** 4

**Summary:**

This paper introduces MetaTT, a Tensor Train (TT)-based adapter framework for parameter-efficient fine-tuning of pre-trained transformers. MetaTT leverages a shared TT decomposition to factorize various transformer sub-modules across structural dimensions such as layer index, matrix type, and optionally heads and tasks. Additionally, they propose a rank-adaptive optimizer inspired by the DMRG method from quantum physics, demonstrating improved optimization when integrated with AdamW for fixed target ranks.

**Strengths:**

The presentation of this paper is clear and easy to follow. The topic of using a tensor-train adapter is quite novel, and the proposed rank-adaptive optimizer appears to be an important and reasonable enhancement that can be effectively integrated with existing tensorized methods.

**Weaknesses:**

- My main concern with this paper is the soundness of the proposed method. I understand that global compression can reduce the number of trainable parameters. However, I’m not convinced why this should lead to better performance. Intuitively, reweighting the entire transformer block should perform worse than reweighting individual linear layers within the block, since adjusting single layers allows more flexibility—especially when combined with in-block non-linearity. I didn’t see any discussion or justification from the authors on why the proposed method works in this regard.
- The experimental results further reinforce my concern. For example, in Table 1, the proposed method shows improvements over baselines like LoRA, but the gain is usually less than 1%, which is not substantial enough to confidently claim a real improvement. A similar trend is observed across other tables.
- I’m also wondering whether the DMRG optimizer is included in Tables 1–3. If not, why was it excluded? Based on the results and discussion, it seems that the DMRG-inspired method itself contributes significantly to the performance gains. In contrast, the improvements from the tensorized adapter alone appear to be limited.
- BERT-based models feel somewhat out-of-date to me. I highly suggest that the authors focus more on the tasks in Table 1 for the ablation studies instead.

**Questions:**

See the weakness

---

> ### Author Response · Authors · 2025-11-26
> **Response to Weaknesses**
>
> We thank the reviewer for their comments. We address the weakness below.
>
> - We agree that the question of whether global compression offers benefits beyond parameter reduction is an important one. Our intuition is that certain weights may be redundant across layers and matrix types, and MetaTT is able to exploit these redundancies, unlike LoRA. Currently, evidence for such redundancies -- and thus for the effectiveness of parameter sharing -- is primarily empirical, as discussed in the works cited in Section 1. Additionally, the universality of the TT decomposition ensures that any tensor-based adapter can, in principle, be represented in TT form, further supporting the flexibility of this approach.
>
> - While our methods either perform comparably or outperform existing methods in terms of accuracy, it does so with a considerably smaller parameter budget (e.g., in Table 1 and Table 2 we see about 2x to 20x parameter reduction, in Table 3 we see about 42x parameter reduction). This is achieved thanks to MetaTT's global compression mechanism. We show that it works well in practice, while providing advantages in multi-task training and rank-adaptive optimization (via DMRG).
>
> - The results in Tables 1 to 3 does not include DMRG and is stated to demonstrate the power of the adapters alone. In Section 2.4, Section 3.3, and Appendix C, we discuss DMRG and its results. We have also included a new figure (Appendix I.2, Figure 13) comparing MetaTT-4D with DMRG against LoRA and AdaLoRA on Commonsense15k (down-sampled from Commonsense170k used in Table 1), showing the inclusion of DMRG-style sweeps boosts signficantly the performance of MetaTT-4D.
>
> - Both RoBERTa base and large models are small enough that can be easily acquired and tested on, including for our MTL and DMRG experiments. Due to short turn around time of the rebuttal and the need to demonstrate a wide array of ablations, we were unable to run additional ablations on Llama based models (except for additional experiments with DMRG which can be found in Appendix I.2). We will include ablation studies on Llama, including multiple random seeds, for our camera ready version.

---

> > ### Comment · Reviewer_6E7j · 2025-11-26
> >
> > Thank you for the authors’ response. After careful consideration, I am still concerned about the effectiveness of the proposed method and the relatively limited experimental improvements. In addition, reducing the number of parameters does not directly translate to real-world efficiency: since a full backpropagation graph is still required, the reduction in training memory may be limited, and the existing tensor contractions may even increase training time. These issues should be evaluated more thoroughly. Thus, I will maintain my score.

---

> ### Author Response · Authors · 2025-11-27
>
> We thank the reviewer for taking the time to review our rebuttal, additional comments, and allowing us the opportunity to improve our work. To alleviate outstanding concerns regarding computational graphs, backprop efficiency, and general runtime, we redirect the reviewer to Section J of our rebuttal text which compares the backward and forward pass across adapters and models. We show them here:
>
> | Model | Adapter | Rank | Batch | Forward pass (secs) | Backward pass (secs) |
> |-------|---------|------|-------|-------------------|---------------------|
> | RoBERTa-base | LoRA | 8 | 64 | **0.1859 (1)** | **0.2014 (1)** |
> | RoBERTa-base | VeRA | 1024 | 64 | 0.2539 (0) | 0.2718 (1) |
> | RoBERTa-base | MetaTT-4D | 24×3 | 64 | **0.1866 (1)** | **0.2031 (0)** |
> | RoBERTa-base | MetaTT-5D | 64×4 | 64 | 0.1921 (1) | 0.2146 (1) |
> | RoBERTa-large | LoRA | 8 | 64 | **0.62 (2)** | **0.6575 (1)** |
> | RoBERTa-large | VeRA | 256 | 64 | 0.6465 (2) | 0.7035 (3) |
> | RoBERTa-large | MetaTT-4D | 32×3 | 64 | **0.6087 (0)** | **0.6612 (1)** |
> | RoBERTa-large | MetaTT-5D | 64×4 | 64 | 0.63 (1) | 0.6917 (2) |
>
> **Table Caption:** World clock runtime comparison of different PEFT adapters. We report the average time required to make 5 batches pass through the model and corresponding adapter and then compute gradients, 5 independent trials. We observe that in general LoRA and MetaTT-4D are the fastest among the other methods reported here for their respective best performing ranks. We note that even for similar number of trainable parameters in VeRA, the matrix-vector-vector-matrix operation is significantly larger than the matrix-matrix operations in LoRA and variants of MetaTT, leading to gains in the forward and backward pass across batch and trials.
>
> We reiterate that **MetaTT does not introduce exotic tensor operations with high overhead** -- it reduces to **standard matrix-matrix multiplications**, much like LoRA. This is evident in the runtimes reported in the table, which show MetaTT achieving comparable speeds to LoRA -- and even slightly outperforming it at inference, likely due to the use of more square matrices in MetaTT (as higher ranks are required compared to LoRA). By design, MetaTT avoids the less efficient "matrix-vector-vector-matrix" patterns that slow down VeRA and the overhead associated with higher order tensor decompositions. This principle is central to MetaTT and directly addresses the first point raised by the reviewer: while single-layer adapters offer more flexibility, their data layouts -- such as those used in vector-based or higher-order tensor adapters -- are typically less efficient for GPU computation.
>
> In the table below, we report the peak GPU memory usage during training for both LoRA and MetaTT adapters on Llama-2-7b, trained on Commonsense15k with a batch size of 8, using a 40GB A100 GPU. Results are shown as a percentage of total GPU memory, averaged over three trials with standard errors in parentheses. As shown, MetaTT-4D consistently achieves a lower memory footprint, with only a modest increase as rank grows, while LoRA exhibits a more pronounced relative increase. These findings demonstrate that MetaTT provides better training memory efficiency than LoRA. We hope this addresses the reviewer’s concerns regarding MetaTT’s training memory usage, and we are happy to provide further memory profiling results upon request.
>
> | Adapter | Rank | memory.used/memory.total (%) |
> |-------|---------|------|
> LoRA | 16 | 86(4) |
> MetaTT-4D | 16x3 | 83(4) |
> MetaTT-5D | 16×4 | 89(3)|
> LoRA | 64 | 94(1) |
> MetaTT-4D | 64x3 | 85(6) |
> MetaTT-5D | 64x4 | 87(5) |

---

### Official Review · Reviewer_73xA · 2025-10-28

**Soundness:** 3
**Presentation:** 2
**Contribution:** 2
**Rating:** 2
**Confidence:** 5

**Summary:**

This paper introduces MetaTT, a parameter-efficient fine-tuning framework based on tensor train decomposition. MetaTT parameterizes all transformer sub-modules using a single shared TT, achieving global compression. The paper proposes two variants, MetaTT-4D and MetaTT-5D, for single-task fine-tuning and extends the architecture to joint multi-task learning with an additional tensor core. Furthermore, the paper leverages a rank-adaptive optimizer inspired by the DMRG method from quantum many-body physics to enhance optimization. Experimental results compare MetaTT against several PEFT baselines, including LoRA, VeRA, and LoTR, for both single-task and multi-task settings on standard benchmarks like GLUE and commonsense reasoning datasets.

**Strengths:**

1. The proposed use of a single shared TT for compressing all transformer layers and sub-modules is novel and shows promise in reducing the parameter count while maintaining competitive performance.
2. The rank-adaptive training inspired by DMRG is an interesting integration of techniques from quantum physics into machine learning, showcasing interdisciplinarity and potential for further exploration.
3. The paper compares MetaTT with several state-of-the-art PEFT methods, including LoRA and LoTR, across multiple tasks, and reports detailed results on parameter efficiency and accuracy.

**Weaknesses:**

1. Despite the novelty of the approach, the reported performance improvements are marginal or absent in most cases compared to simpler baselines like LoRA, especially given the significant computational complexity added by TT decomposition and rank-adaptive training.
2. While the DMRG-inspired optimizer is presented as a key contribution, its practical benefits over standard optimizers like AdamW are not convincingly demonstrated. The rank-adaptive approach introduces additional training complexity without a clear payoff in terms of accuracy or efficiency.
3. The paper overlooks some recent works on tensor-based adapters and PEFT methods, particularly those focusing on computational trade-offs and scalability, such as AdaLoRA and other adaptive rank methods.
4. While the authors provide detailed pseudocode, the implementation lacks clarity in critical areas like initialization strategies and hyperparameter choices, which are shown to influence MetaTT's performance heavily. This raises concerns about the reproducibility of results.

**Questions:**

1. The results show that MetaTT performs similarly to LoRA and LoTR in many benchmarks, with only marginal improvements in some cases. Can the authors clarify the practical advantages of MetaTT over these simpler methods, particularly in real-world scenarios? How do the authors justify the significant computational overhead introduced by TT decomposition and rank-adaptive training in light of these modest gains?
2. While the paper provides pseudocode and hyperparameter grids, the results seem highly dependent on initialization strategies and specific rank settings. Could the authors share more details about the exact initialization methods, hyperparameter tuning process, and any challenges encountered during experimentation? Are there plans to release the full implementation and training pipelines?
3. The paper notes that MetaTT-5D is more sensitive to initialization and training instability than MetaTT-4D. Can the authors elaborate on why this is the case? Are there specific guidelines or heuristics for initialization and hyperparameter selection that can make MetaTT-5D more robust? How does this sensitivity impact the usability of MetaTT in practice?

---

> ### Author Response · Authors · 2025-11-26
> **Response to Weaknesses**
>
> We thank the reviewer for their time in reviewing our work, and for the helpful comments.
>
> 1. We believe our performance improvements to be significant in light of the parameter reduction. We point the reviewer's attention to Table 1 and Table 2, where we find comparable performance with 2x to 20x parameter compression agaisnt LoRA, or Table 3 where we witness a 42x parameter reduction, in the multi-task setting.
>
> 2. We point the reviewer to Figure 2 of the main text for a comparison of optimizing MetaTT using AdamW vs. AdamW + Algorithm 1 (our proposed rank adaptive scheme) when fine-tuning on RoBERTa. We have also added a new figure in Appendix I.2 in the revised version, where we extend this analysis to Llama-2 7b on Commonsense15k (a downsampled version of Commonsense170k). The results demonstrate that MetaTT-4D trained with DMRG-inspired sweeps not only surpasses the version trained with AdamW alone at target rank $r=20$, but also outperforms other higher-rank variants. Furthermore, our comparison shows that MetaTT-4D exceeds the performance of both LoRA and AdaLoRA variants, achieving accuracy on par with the best-performing LoRA adapter at rank 16, while using approximately 47 times fewer parameters. We remark that the overhead of Algorithm 1 is minimal when compared to AdamW. Instead there is a slight overhead stemming from Pytorch's need to recompile the computational graph and reinitialize CUDA kernels after the adapter shapes are modified. In Table 18 we show the times required to train with LoRA, AdaLoRA, MetaTT-4D, and MetaTT-4D+DMRG, showing the small relative overhead incurred from this artifact.
>
> 3. We discussed AdaLoRA implicitly in the introduction and in Appendix C, acknowledging its relevance to our work. As noted in Appendix C, AdaLoRA avoids performing SVDs across all layers precisely because this would be computationally prohibitive. Instead, AdaLoRA proposes a pruning strategy, where a proxy score is used to approximate the importance of singular values, thereby circumventing the need for explicit SVD computations. In Appendix I, we provide a direct comparison between AdaLoRA and LoRA, observing that the improvements offered by AdaLoRA over LoRA are often marginal. Furthermore, Figure 13 compares the performance of AdaLoRA with MetaTT+DMRG, demonstrating that MetaTT with DMRG not only outperforms MetaTT variants trained with AdamW alone, but it also performs competitively against LoRA variants with substantially fewer parameters. Additionally, MetaTT adapters store matrices in dense format -- unlike AdaLoRA, which relies on pruning -- making them more suitable for efficient GPU inference after training.
>
> 4. We had reported the hyper-parameters chosen to report the results in our work in Appendix D. We had also included ablations on adapter initialization in Appendix D.4. Furthermore, we have added ablations on a wider set of seeds to study stability of MetaTT variants, LoRA, VeRA in Appendix H. We hope that until our code is made public, these would sufficiently satisfy any concerns regarding reproduciblity of our work.

---

> ### Author Response · Authors · 2025-11-26
> **Response to Question 1**
>
> 1. Regarding computational overhead -- we re-emphasize, the tensor train can be written as matrix products, and so can be very efficiently computed in GPUs. To compare world clock executing times, we added a section in the appendix comparing LoRA, VeRA, MetaTT-4D, and MetaTT-5D (Appendix J). For completeness we add the numbers here.
>
> | **Model** | **Adapter** | **Rank** | **Batch** | **Forward pass (secs)** | **Backward pass (secs)** |
> |-----------|-------------|----------|-----------|-------------------------|---------------------------|
> | RoBERTa-base | LoRA | 8 | 64 | **0.1859 (1)** | **0.2014(1)** |
> | RoBERTa-base | VeRA | 1024 | 64 | 0.2539(0) | 0.2718(1) |
> | RoBERTa-base | MetaTT-4D | 24×3 | 64 | **0.1866(1)** | **0.2031(0)** |
> | RoBERTa-base | MetaTT-5D | 64×4 | 64 | 0.1921(1) | 0.2146(1) |
> | RoBERTa-large | LoRA | 8 | 64 | **0.62(2)** | **0.6575(1)** |
> | RoBERTa-large | VeRA | 256 | 64 | 0.6465(2) | 0.7035(3) |
> | RoBERTa-large | MetaTT-4D | 32×3 | 64 | **0.6087(0)** | **0.6612(1)** |
> | RoBERTa-large | MetaTT-5D | 64×4 | 64 | 0.63(1) | 0.6917(2) |
>
> **Caption:** World clock runtime comparison of different PEFT adapters. We report the average time required to make 5 batches pass through the model and corresponding adapter and then compute gradients, 5 independent trials. We observe that in general LoRA and MetaTT-4D are the fastest among the other methods reported here.
>
> Finally we would like to reinforce, that even beyond computational benefits, in practical scenarios, the MetaTT adapter is considerably smaller in memory than its existing counterparts, making it easier to store and rapidly change in dynamic tasks. It natively supports rank-adaptive optimization, which benefits from existing theoretical and empirical explorations, as well as our own demonstrations of competitiveness in training language model architectures. Beyond single task learning, MetaTT adapters can be easily applied to multi-task learning, which allows for an even further parameter compression.

---

> ### Author Response · Authors · 2025-11-26
> **Response to Question 2**
>
> 2. We agree with the reviewer on the importance of reproducibility and have shared python code (not just pseudocode), that can be used to replicate our experiments (see Appendix E). Moreover, we believe to have shared details that the reviewer asks for:
>     - in Appendix D, we have detailed the exact set of hyper-parameters across methods, including MetaTT used to reproduce the results presented in our submission.
>     - in Appendix D.4 we have discussed how performance of MetaTT changes as we use different initialization schemes. We observe that the adapter performance doesn't significantly change across initialization. In spirit, this is similar to how LoRA adapter behave when the Gaussian initialization is swapped for initializing with singlar vectors of the pre-trained weights (Pissa, OLoRA). We add a small discussion in Appendix H.4 on this.
>     - We also have intentions to share our code on a GitHub repository in the near future. However, we are unable to do so at present.

---

> ### Author Response · Authors · 2025-11-26
> **Response to Question 3**
>
> 3. As a response to the reviewer's concerns on training stability, we have run updated experiments comparing the sensitivity to initialization in Appendix H, when using LoRA, VeRA, MetaTT-4D, and MetaTT-5D in several experimental regimes.  We observe the following:
>     - When the hyper-parameters are fixed (and chosen according to the ones reported in Section D) we compare the half width of the 95\% confidence interval across $8$ independent trials (also called margin of error). We observe that across CoLA, MRPC, and RTE and Roberta-base and Roberta-large, the margin of error remains similar. In Figure 6 we also demonstrate the variance on the validation accuracy during training epochs. We observe that the variants of MetaTT perform similarly when compared to other adapters.
>     - Keeping every other hyper-parameters constant when only the learning rates are varied, we observe that MetaTT-4D behaves similar to LoRA across all tasks compared both Roberta-base and Roberta-large. We also observe that MetaTT-5D behaves similar to VeRA on Roberta-base.
>     - When other hyper-parameters are kept constant and only rank is varied, MetaTT-4D becomes worse with increasing rank and MetaTT-5D improves with increasing rank. However, one should note that for LoRA both rank and $\alpha$ was varried ($\alpha$ is known to be 2*rank for LoRA) and VeRA handles alpha implicitly. Furthermmore, for MRPC and RTE, on Roberta-large, all adapters behave similarly when ran is varied. We belive this is a stronger evidence to the model capacity induced by these TT variants over LoRA and VeRA.. We expect that the decreased robustness in training may be an function of the increased parameter sharing: since a parameter in MetaTT-5D affects a larger number of weights across the architecture compared to MetaTT-4D. However, deeper theoretical considerations are left to future work.
>
> We believe all these experiments confirm that MetaTT adapters can be very easily used as a replacement for LoRA like other PEFT adapters without any issues. We posit that the amount of effort required to find the right set of hyper-parameters for a specific task is no harder than finding the right set of hyper-parameters for other PEFT adapters.

---

### Official Review · Reviewer_Kszv · 2025-10-31

**Soundness:** 3
**Presentation:** 3
**Contribution:** 3
**Rating:** 8
**Confidence:** 3

**Summary:**

The submission introduces a Tensor Train adapter model that parameterizes LLM weight updates as a fourth or fifth order tensor. Moreover, they introduce an optimization scheme that starts with a larger rank and iteratively fits an adapter and applies (approximated) truncated SVD to reduce the rank.

**Strengths:**

The paper is well written, easy to follow, the contribution and references to prior work are clear, the approach is sound and experiments not only include the standard benchmarks but illustrate particularities of their proposed methods.

Both TT parameterizations (e.g. LoTR) and fifth order tensor adapter models that use layers input output dimensions and heads  (e.g. LoRTA) have been proposed, but not their conjunction. Secondly, treating tasks as an additional dimension is, to the best of my knowledge, novel.

The iterative DMRG inspired training algorithm is novel and relevant contribution that motivates further research into designing optimization algorithms for low rank tensor adapters that dynamically adjust rank throughout optimization. Although truncated SVD has been proposed to initialise LoRA adapters, and some optimizers have been proposed specifically for low rank matrix adapters (e.g. GaLoRE), the low rank tensor literature primarily relies on standard (adamw) optimization tools and, more importantly, the rank is usually treated as a fixed hyper-parameter. This is relevant because when the rank is low - regardless of the adapter model - optimization dynamics usually becomes challenging and starting with a larger rank can mitigate this.

**Weaknesses:**

I think that the rank adaptive optimization scheme is a strong contribution. The experiments that showcase its benefits are centered in the standard NLU setting with roberta, but I think it would be useful to extend the empirical analysis to (at least one, ideally all) of other benchmarks/tasks/models in order to further substantiate the empirical gains from this scheme in settings that are regarded as more challenging.

**Questions:**

Why do most experiments show only meta-TT 4D (except for roberta in nlu tasks)?

Can you comment on initialization and sensitivity - perhaps provide an ablation ?

---

> ### Author Response · Authors · 2025-11-26
>
> We thank the reviewer for the time taken to review our work and their helpful comments.
>
> In response to the review, we have included further experimental evidence of the effectiveness of DMRG. We have also included a new figure in Appendix I.2 where we extend the application of DMRG to Llama-2 7b on Commonsense15k (a downsampled version of Commonsense170k), showing that MetaTT-4D trained with DMRG-inspired sweeps not only outperforms the version trained with AdamW alone for a target rank $r=20$, but also other variants of higher rank as well as LoRA and AdaLoRA variants.
>
> Experimentally, we found MetaTT-5D struggled more than MetaTT-4D in Commonsense170k and so we excluded these results in our preliminary submission. One possibility is that MetaTT-5D is more sensitive to learning rates and $\alpha$ than its 4D counterpart on Llama-2. We hope to get a better understanding of why this is by the final submission of our work.
>
> Our novel adapters have comparable sensitivity to hyper-parameters to the existing adapters. To illustrate this, in Appendix H, we have included a detailed analysis of sensitivity of LoRA, VeRA, MetaTT-4D, and MetaTT-5D in several experimental regimes.
>
>  - When the hyper-parameters are fixed (and chosen according to the ones reported in Section D) we compare the half width of the 95\% confidence interval across $8$ independent trials (also called margin of error). We observe that across CoLA, MRPC, and RTE and Roberta-base and Roberta-large, the margin of error remains similar. In Figure 6 we also demonstrate the variance on the validation accuracy during training epochs. We observe that the variants off MetaTT perform similarly when compared to other adapters.
>
>  - Keeping every other hyper-parameters constant when only the learning rates are varied, we observe that MetaTT-4D behaves similar to LoRA across all tasks compared both Roberta-base and Roberta-large. We also observe that MetaTT-5D behaves similar to VeRA on Roberta-base.
>
>  - When other hyper-parameters are kept constant and only rank is varied, MetaTT-4D becomes worse with increasing rank and MetaTT-5D improves with increasing rank. However, one should note that for LoRA both rank and $\alpha$ was varried ($\alpha$ is known to be 2*rank for LoRA) and VeRA handles alpha implicitly. Furthermmore, for MRPC and RTE, on Roberta-large, all adapters behave similarly when rank is varied. We believe this is a stronger evidence to the model capacity induced by these TT variants over LoRA and VeRA.
>
> In Appendix D.4, we had included the ablations on initializations for MetaTT on MRPC and RTE. We observe that across initializations the adapters behave similarly. We also include a small section on the different initializations of LoRA in Appendix H.4. We observe that variants like Pissa, Gaussian, or OLoRA perform very similarly. Unfortunately techniques like using singular vectors of the underlying weights to initialize the adapters does not translate directly to tensors (since there is no one single best decomposition of a tensor) and remains open even in the tensor decomposition community.

---

### Official Review · Reviewer_ufrS · 2025-11-06

**Soundness:** 4
**Presentation:** 4
**Contribution:** 3
**Rating:** 6
**Confidence:** 4

**Summary:**

This paper proposes to use tensor-train factorisation to compress the weight matrices of PEFT models.  They show how this elegant framework can be used to share implicit structure in the weights across parameter types, layers, attention heads, and multiple tasks.  Strong reductions in the number of parameters are achieved while getting similar or better accuracies across a number of tasks.

**Strengths:**

This is an elegant unified approach to compressing PEFT matrices to reduce parameter counts.  It also elegantly extends to multi-task PEFT, allowing shared structure across tasks.

Empirical evaluations are done on a good variety of tasks, using three versions of the model, each an extension of the previous model.  Results are generally good or comparable to previous methods, but with greatly reduced parameter counts.

**Weaknesses:**

The novelty is not high.  There has been a lot of work on PEFT already, and this work does not add much conceptual or theoretical novelty.  The contribution is in identifying a general-purpose mathematical framework which addresses PEFT in a consistent way, rather than a collection of ad-hoc methods.

The empirical results do not demonstrate any breakthroughs with respect to previous work.

**Questions:**

There is earlier work than the papers you cite which seems directly relevant to your approach of factorising parameter matrices and multi-task PEFT, respectively:
 Mahabadi, Henderson, and Ruder. Compacter: Efficient Low-Rank Hypercomplex Adapter Layers.  NeurIPS 2021.
 Mahabadi, Ruder, Dehghani, and Henderson.  Parameter-efficient Multi-task Fintuning for Transformers via Shared Hypernetworks.  ACL 2021.

---

> ### Author Response · Authors · 2025-11-26
>
> We thank the reviewer for taking the time to review our work and their helpful comments.
>
> We agree that there has been a huge body of work on PEFT, our work somewhat tries to address global compressibility using the TT architecture. As can be seen from our experiments in Section 3.1 of our submission, MetaTT variants shows promise in achieving state-of-the-art results while significantly reducing trainable parameters (somewhat similar to other tensor based compression algorithms).
>
> However, our work goes a couple steps beyond just single task fine-tuning. Specifically, in our original and revised submission, we demonstrate that these architectures can be trivially extended to multi-task learning (see Sections 3.2, Appendix B, and Appendix G), and such an extension works reliably when compared to the state-of-the-art methods while reducing the required number of trainable parameters significantly. Furthermore, we also study a rank adaptive scheme inspired by DMRG (see Section 3.3 and Appendix C) that helps to dynamically reduce rank of the adapter during training.
>
> Finally, the adapters used in Compacter and multi-task finetuning via shared hyper-networks are considerably different from our method and we believe they warrant a comparison. So we have added some these references in Section 1.

---

### Author Response · Authors · 2025-11-26
**Updated Version**

We thank all the reviewers for their thoughtful and constructive feedback! We have added the following sections to the original manuscript to address the comments of the reviewers:

 H. Stability Analysis of MetaTT

 I. AdaLoRA vs. MetaTT with DMRG-inspired sweeps

J. Comparison of Runtimes

We have also made some minor updates in the main text, reflected in red. We hope the reviewers consider reflecting their scores considering these updates. We once again thank the reviewers and the AC for their time in reviewing our manuscript.

---

### Meta-Review · Area_Chair_3STt · 2025-12-28

**Summary:**

- Reviewer ufrS suggested the paper's novelty is limited with respect to other PEFT work.
- Reviewer Kszv thinks the contribution is strong in this paper and suggest extending the analysis to other benchmarks.
- Reviewer 73xA thinks the idea is interesting, and raise questions about the marginal improvements, result not being convincing and concern regarding reproducibility.
- Reviewer 6E7j says the paper is easy to follow but was concerned about the soundness of the method.

During rebuttal, the authors tried to addressed a few issues including adding runtime, and memory usage.

**Reviewer Concerns:**

What's addressed:
- The authors addressed the performance issue by showing the huge parameter compression ratio by the method.
- The authors added some discussions regarding applying DMRG to Llama-2 7b on Commonsense15k.

What's not addressed:
- Novelty is still not thorough justified and the discussion on other benchmarks seem limited.
- The model mainly used in the paper is a bit outdated, limiting the overall contribution of this work.

**Reviewer Scores:**

Based on the rebuttal, I think reviewers will likely keep their scores.

---

### Decision · Program_Chairs · 2026-01-26

Reject